# Comparative genomics reveals the dynamics of chromosome evolution in Lepidoptera

Charlotte J. Wright [1] ✉, Lewis Stevens [1], Alexander Mackintosh [2], Mara Lawniczak [1] & Mark Blaxter [1] ✉

Chromosomes are a central unit of genome organization. One-tenth of all described species on Earth are butterflies and moths, the Lepidoptera, which generally possess 31 chromosomes. However, some species display dramatic variation in chromosome number. Here we analyse 210 chromosomally complete lepidopteran genomes and show that the chromosomes of extant lepidopterans are derived from 32 ancestral linkage groups, which we term Merian elements. Merian elements have remained largely intact through 250 million years of evolution and diversification. Against this stable background, eight lineages have undergone extensive reorganization either through numerous fissions or a combination of fusion and fission events. Outside these lineages, fusions are rare and fissions are rarer still. Fusions often involve small, repeat-rich Merian elements and the sex-linked element. Our results reveal the constraints on genome architecture in Lepidoptera and provide a deeper understanding of chromosomal rearrangements in eukaryotic genome evolution.

Chromosomes are the central units of genome architecture in eukaryotic organisms. They determine processes such as recombination and segregation. While chromosomes are generally stable over evolutionary time, large-scale rearrangements, such as fusions and fissions, can occur. Consequently, chromosomes of extant species can be used to infer the linkage groups present in a common ancestor, termed ancestral linkage groups (ALGs). ALGs have been identified in many taxa including Diptera[1], flowering plants[2], Nematoda[3,4], mammals[5], vertebrates[6] and Metazoa[7]. Chromosomal rearrangements have important consequences for genome function[8], speciation[9] and adaptation[10]. For example, heterozygous chromosomal fusions can interfere with meiosis, resulting in reproductively isolated populations[11,12]. The evolutionary forces constraining chromosome number and maintaining ALGs remain unclear. Moreover, how and why certain taxa evade such constraints and experience high rates of karyotypic change are not understood.

In monocentric chromosomes, a single region, the centromere, serves as the organizing centre for Mendelian partitioning of homologues during mitosis and meiosis. Discrete centromeres are absent in holocentric chromosomes as centromeric functions are dispersed along the chromosome. Holocentricity has evolved independently several times across the tree of life, including in nematodes, four times in plants and multiple times in arthropods[13–18]. The most speciose of these holocentric groups is Amphiesmenoptera, comprising the insect orders Lepidoptera (moths and butterflies) and Trichoptera (caddisflies), which together account for 15% of all described eukaryotic species[19,20]. The convergent evolution of holocentricity in many speciose groups indicates that this alternative solution to accurate segregation of chromosomes may be evolutionarily advantageous.

Holocentric chromosomes are suggested to facilitate rapid karyotypic evolution as fragments derived from fission could maintain kinetochore function[21,22]. Lepidoptera are the most karyotypically diverse group of any non-polyploid eukaryote, with haploid chromosome numbers (hereafter chromosome number, *n*) ranging from 5 to 223 (refs. 23,24). However, most species have haploid counts of *n* = 29–31 (refs. 25,26), indicating that further mechanisms must constrain holocentric karyotype evolution. Indeed, chromosome numbers and their gene contents are generally stable over evolutionary time in both holocentric and monocentric taxa[27].

Changes in chromosome number alter the recombination rate[28,29]. In Lepidoptera, where recombination only occurs in males (ZZ), there

[1]Tree of Life, Wellcome Sanger Institute, Cambridge, UK. [2]Institute of Ecology and Evolution, University of Edinburgh, Edinburgh, UK. ✉e-mail: charlotte.wright@sanger.ac.uk; mark.blaxter@sanger.ac.uk

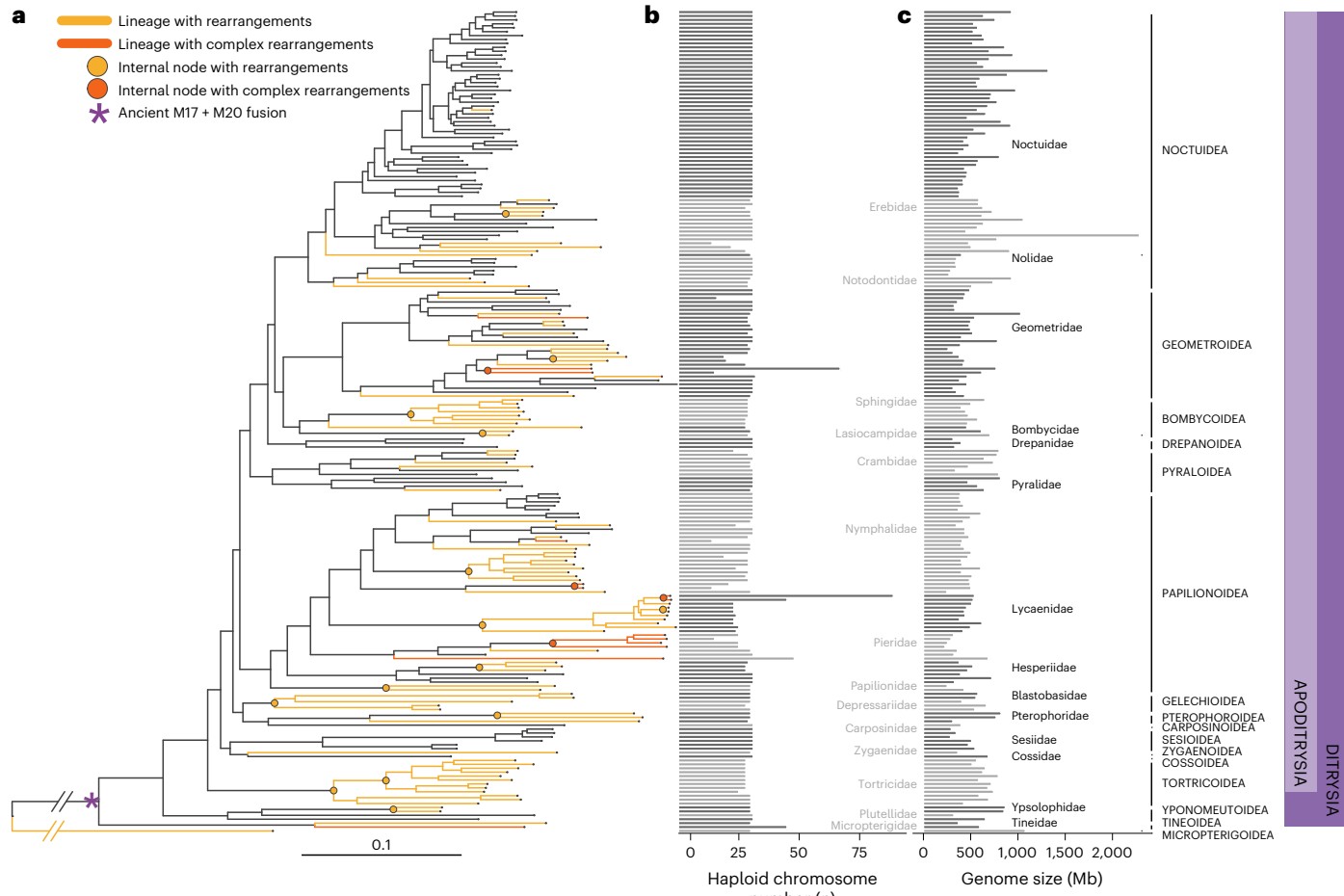

**Fig. 1 | Phylogenetic relationships of 210 lepidopteran species and the distribution of large-scale rearrangement events. a**, Phylogeny was inferred using the amino acid sequences of 4,947 orthologues that were present and single copy in 90% of all species sampled under the LG substitution model with gamma-distributed rate variation among sites. The tree was rooted using five representative species of the two main suborders from the sister group, Trichoptera (caddisflies). Excluding the ancient fusion between M17 and M20, which is shared by all Ditrysians (purple asterisk), half of the species have retained intact Merian elements since the last common ancestor of Lepidoptera (black lines). Orange branches indicate lineages with at least one fusion or fission event. Orange circles indicate internal nodes where descendants share a fusion event. We inferred no fission events at internal orange nodes. Red branches indicate lineages with extensively reorganized genomes (*Lysandra coridon, Lysandra bellargus, Pieris brassicae, Pieris napi, Pieris rapae, Tinea semifulvella, Melinaea menophilus, Melinaea marsaeus, Aporia crataegi, Brenthis ino, Operophtera brumata, Phylereme vetulata, Leptidea sinapis* and *Apeira syringaria*). Red nodes indicate internal nodes where extensively reorganized descendants share fusion or fission events. Scale in substitutions per site is shown. **b,c**, The distribution of haploid chromosome number (*n*) (**b**) and genome size (Mb) (**c**) across 210 lepidopteran species. Alternating shades distinguish different taxonomic families. Source data for this figure can be found in Supplementary Tables 1 and 6 and in the Zenodo repository[122].

tends to be one crossover event per chromosome per generation[30–32]. Thus, loci on a fused chromosome formed from two equally sized progenitors will experience a 50% reduction in per base recombination rate relative to the unfused chromosomes. Changes in recombination rate will impact the evolutionary forces that shape genome architecture, altering the effect of selection at linked sites and therefore effective population size. Lower recombination rates also intensify Hill–Robertson interference between tightly linked beneficial loci, hindering adaptive evolution[33]. However, local adaptation is facilitated by reduced recombination between locally adapted loci in the presence of gene flow[34,35].

Here, we infer ALGs for Lepidoptera, which we term Merian elements, from 210 chromosomal genome assemblies using a reference-free, phylogenetically aware approach. We find that Merian elements have remained intact in most species. While infrequent fusions occur, fissions are extremely rare. Constraints on large-scale reorganization have been relaxed in eight lineages, resulting in chromosomes that are the products of either many fissions or numerous fusion and fission events. Across Lepidoptera, we find that fusions are biased towards shorter autosomes and the Z sex chromosome, suggesting that both chromosome length and haploidy in the heterogametic sex play key roles in constraining genome rearrangement.

## Over 200 chromosomally complete lepidopteran genomes

To explore karyotype variation across Lepidoptera, we selected chromosome-level reference genomes for 210 species of Lepidoptera, representing 16 of the 43 (37%) superfamilies, including basal lineages such as Micropterigidae and Tineidae. Almost 90% of the assemblies (188 of 210) were generated by the Darwin Tree of Life project[36] (Supplementary Table 1). These reference genomes are high-quality, with high gene completeness (mean 98.24%, s.d. = 1.75%); assessed by benchmarking using single-copy orthologues (BUSCO; lepidoptera odb10 dataset)[37], high contiguity (mean contig N50 13.47 Mb, s.d. = 6.92) and the vast majority of each assembly scaffolded into chromosomes (mean 99.56%, s.d. = 1.28) (Supplementary Tables 1, 2 and 3 and Supplementary Figs. 1 and 2). Using BUSCO loci, we inferred a phylogeny of the 210 species, which we rooted with five Trichoptera (caddisflies; Fig. 1a).

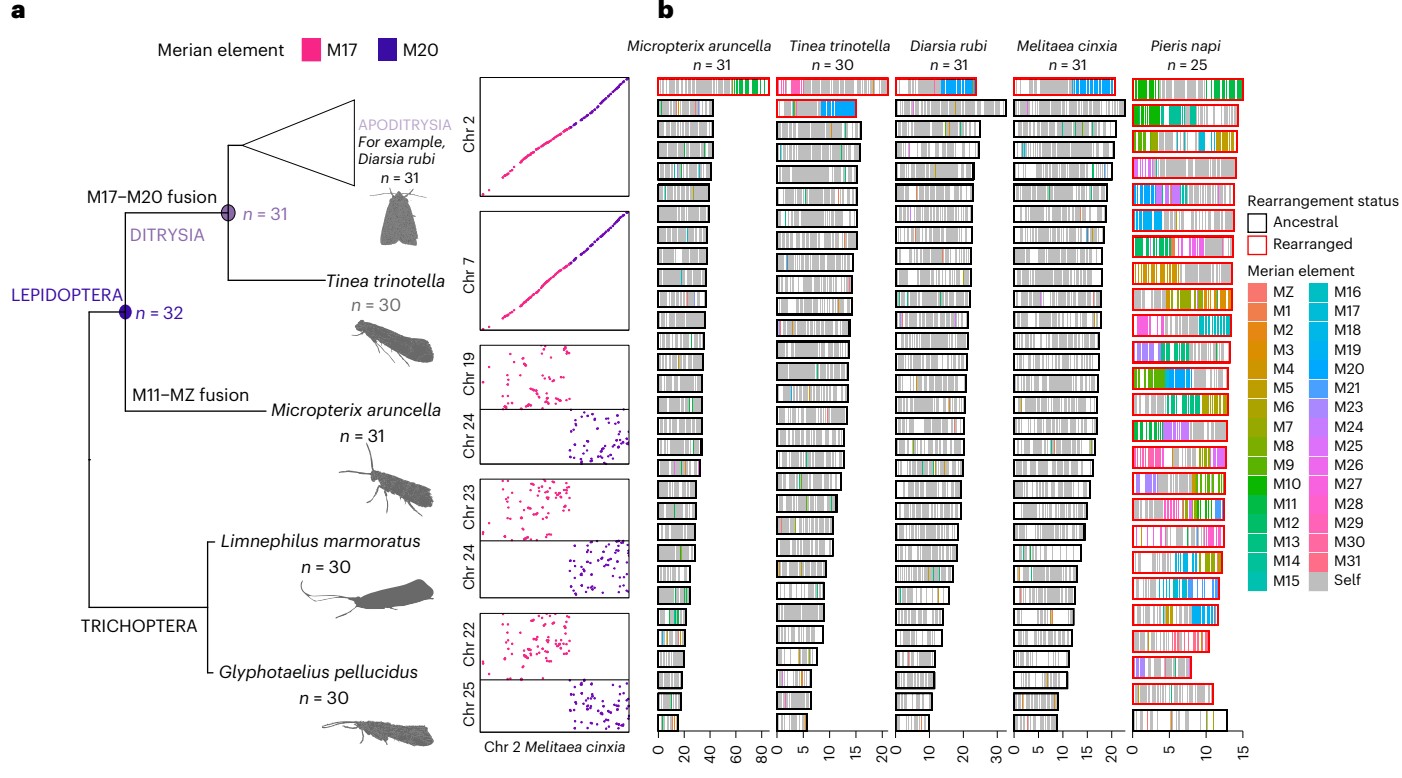

**Fig. 2 | Defining 32 Merian elements. a**, Inferred ancestral karyotype of Lepidoptera and the fusion between M17 and M20 found in all Ditrysia. The phylogeny contains representatives of Trichoptera, *Limnephilus marmoratus* and *Glyphotaelius pellucidus*, in addition to the early-diverging lineage within Lepidoptera, *Micropterix aruncella* and the early-diverging lineage within Ditrysia, *Tinea trinotella* and a representative of *Ditrysia, Diarsia rubi*. To the right of each species in the phylogeny, an Oxford plot of the chromosomes containing orthologues belonging to M17 and M20 in the species is shown relative to *Melitaea cinxia*, which has the chromosome complements of a typical ditrysian species. **b**, Merian elements painted across the chromosomes of *Micropterix aruncella, Tinea trinotella, Diarsia rubi, Melitaea cinxia* and *Pieris napi*. Each

chromosome is represented by a rectangle within which the position of each orthologue is painted grey if it belongs to the most common Merian element for that chromosome or else coloured by the alternative Merian element. Chromosomes that have undergone fusions and/or fission events are outlined in red. Source data for this figure can be found in Supplementary Tables 4 and 10, the Zenodo repository[122] and in the Source Data. Silhouette of *Limnephilus lunatus* by Christoph Schomburg, PhyloPic. Credits for the photographs from which the remaining silhouettes were derived: *Diarsia rubi* and *Glyphotaelius pellucidus*, Donald Hobern/Flickr; *Tinea trinotella*, Ilia Ustyantsev/Flickr; *Micropterix aruncella*, Christoph Schomburg/Flickr; all adapted under a Creative Commons license CC BY-SA 2.0 DEED.

The karyotypes inferred from the genome assemblies were consistent with previous cytological determinations, ranging from $n = 14$ in *Brenthis ino* to $n = 90$ in *Lysandra coridon*[26]. Four-fifths (82%) of the lepidopteran species had an assembled $n$ of 28–31 (Fig. 1b and Supplementary Fig. 3). Genome size varied tenfold, from 230 Mb (*Aporia crataegi*) to 2.29 Gb (*Euclidia mi*) (Fig. 1c and Supplementary Fig. 4). In contrast to previous studies[38], we found no significant correlation between genome size and chromosome number (phylogenetic linear model, $t = 0.83$, $P = 0.4087$, adjusted $r^2 = 0.00795$). (Supplementary Fig. 5).

We observed strong patterning of features along each chromosome, including GC content, repeat and coding densities, consistent with previous observations[39]. Both GC content and repeat density were higher towards the ends of chromosomes compared to their centres (Extended Data Fig. 1a,b). In contrast, coding density tended to decrease towards chromosome ends (Extended Data Fig. 1c). Normalizing for chromosome length, we found that the pattern of feature distribution was similar across all autosomes and the Z chromosome (Extended Data Fig. 1d).

## Thirty-two ancestral lepidopteran linkage groups

We used 5,287 single-copy orthologues in 210 lepidopteran and 4 trichopteran species to define ALGs in a reference-free, phylogenetically aware manner (Fig. 1a), using the tool syngraph[40]. In brief,

syngraph implements an adjacency-based approach which exploits the co-occurrence of loci on the same chromosome, without regard to their order, to infer linkage groups and interchromosomal rearrangements. Although previous work proposed 31 ALGs in the last common ancestor of Lepidoptera[41–43], we assigned 4,112 orthologues (78%) to 32 ALGs (Fig. 2a): 31 autosomes and Z, the sex chromosome. Hereafter, we refer to these ALGs as Merian elements, named after the seventeenth-century lepidopterist and botanical artist, Maria Sibylla Merian[44]. Merian elements were named in order of the number of orthologues they carry, ranging from 273 in the largest Merian element (M1) to 19 in the smallest (M31). The sex-linked Merian element (MZ) contains 161 orthologues (Supplementary Table 4). We tested the robustness of syngraph inferences by performing 100 bootstrap replicates and consistently recovered the same 32 ALGs (Methods). The independent ALG inference tool AGORA[45] yielded highly congruent results (Supplementary Text and Supplementary Figs. 6 and 7).

An ancient fusion involving M17 and M20 occurred on the branch leading to the last common ancestor of Ditrysia, the most taxonomically and ecologically diverse group of Lepidoptera (Fig. 1a), generating the 31 linkage groups observed in most extant Ditrysia. We refer to this fusion as 'M17 + M20', where the '+' denotes an end-to-end fusion, without mixing of genes. In *Micropterix aruncella*, from the early-branching family Micropterigidae, M17 and M20 are distinct chromosomes. M17 and M20 ALGs were also distinct in the last common ancestor

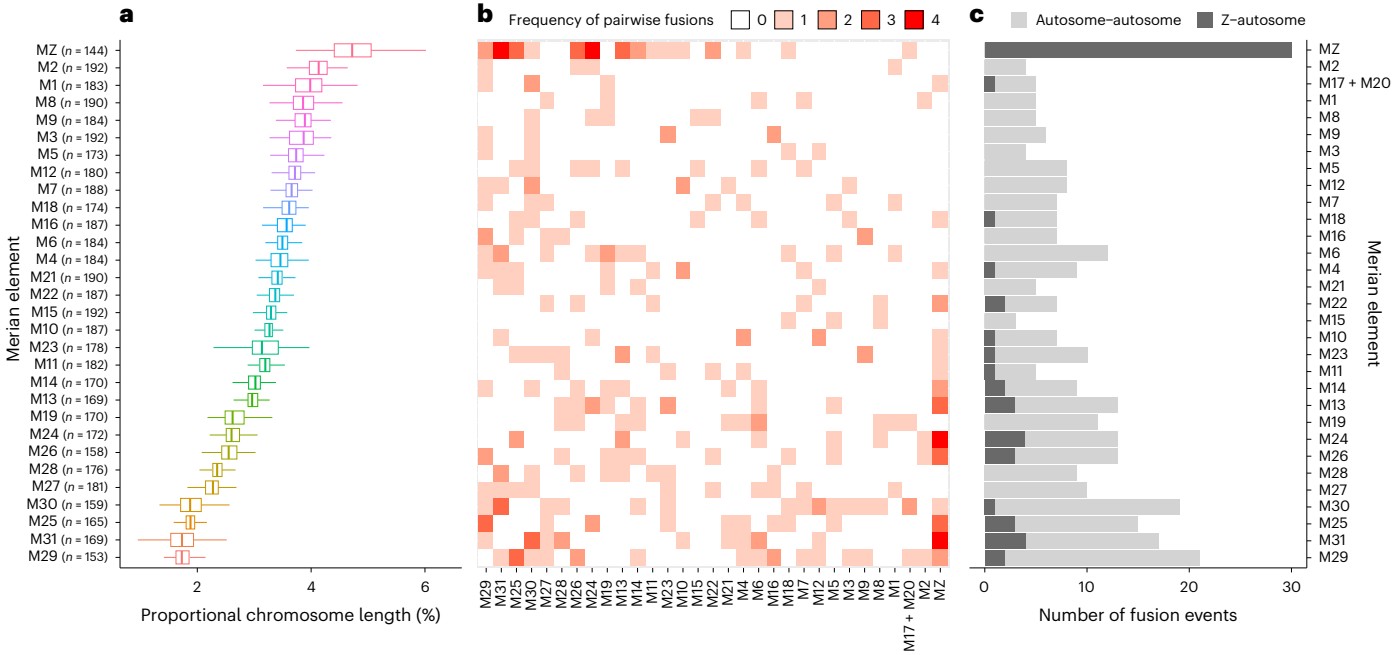

**Fig. 3 | The relationship between Merian element length and tendency to be involved in fusions. a**, Conservation of Merian element length across Lepidoptera. Box plots of the variation in proportional chromosome length within each Merian element. The box plots show the median (centre line) and the first and third quartiles (Q1 and Q3; box limits) and the whiskers extend to the last point within 1.5 times the interquartile range below and above Q1 and Q3 respectively. Observations that fall outside Q1 and Q3 are shown as outliers. Only Merian elements that have remained intact (no large-scale rearrangements) were included. **b**, Matrix of fusion events between pairs of Merian elements, where the shade of red indicates the total number of fusion events per Merian element. **c**, Bar chart of the number of autosome–autosome and sex chromosome–autosome fusion events that each Merian element was involved in. Merian elements are ordered on the basis of average proportional length across the 210 species. Source data for this figure can be found in Supplementary Tables 1, 6 and 10 and in Source data.

of Trichoptera. As the separations of loci defining M17 and M20 were identical in *M. aruncella* and the four Trichoptera, this excludes the possibility that these represent two independent fissions of an ancestral element (Fig. 2a).

We explored the evolutionary dynamics of Merian elements by 'painting' the positions of the orthologues that define each element onto chromosomes of present-day species (Fig. 2b). Except for the ancient M17 + M20 fusion, the chromosomes of most species corresponded to intact Merian elements. Simple fusion and fission events identified in several species reflected previous cytological karyotype assessments[26]. For example, the chromosomes of *M. aruncella* directly corresponded to single, intact Merian elements, with the exception of one Z–autosome fusion (MZ + M11). We identified a distinct Z–autosome fusion (MZ + M29) in *Tinea trinotella* which is consistent with a cytological *n* of 30 (ref. 46). Gene order synteny within each element was highly conserved, even after chromosomal fusion events, including the ancient M17 + M20 (Fig. 2a). More complex rearrangements have occurred in 14 species from 8 lineages. For example, in *Pieris napi*, most chromosomes were made up of segments derived from more than one Merian element and individual Merian elements were fragmented across several chromosomes, indicating a history of many fusion and fission events, as proposed previously[47]. In chromosomes that had not undergone rearrangement events, the proportional length of each Merian element was broadly conserved across species (Fig. 3a). We compared the distribution of the orthologues allocated to Merian elements to their allocation to bilaterian ALGs (BLGs; *n* = 24)[7], from which Merian elements descend and which date to ~560 million years ago[48]. As expected, Merian elements show some correspondence to BLGs with 17 Merian elements showing greater similarity in orthologue assignment with BLGs than expected under random sampling. However, most Merian elements were rearranged relative to BLGs, possessing combinations of loci from multiple BLGs (Extended Data Fig. 2a–c).

## Distribution of fusion and fission events across Lepidoptera

Merian elements provide a foundation for the inference of pattern and process in lepidopteran chromosome evolution. We used phylogenetically aware tools to infer the rearrangement histories of 196 species where chromosome painting indicated simple fusions between complete Merian elements or fission of single Merian elements.

Excluding the ancient M17 + M20 fusion, 54% (106 of 196 species) have retained intact Merian elements since the last common ancestor of Lepidoptera. In the 90 Ditrysian species that deviate from *n* = 31, we identified 183 simple fusion events and four fission events (Fig. 1b). Fission was observed in just three species (*Celastrina argiolus*, *Macaria notata* and *Eupithecia centaureata*), which have one, one and two fissions, respectively. We also identified a single instance where segments of two Merian elements had fused together and the remaining portions existed as separate chromosomes, resulting from two fissions (M1 and M6 in *Eupithecia centaureata*) (Supplementary Fig. 8). Most (159, 86%) of the 183 simple fusions appeared to be evolutionarily young, as they were observed in single species. However, 25 fusions mapped to 14 internal nodes and were shared by all descendants (Fig. 1a) (Supplementary Tables 5 and 6). In all fusions, the domains derived from the ancestral chromosomes remained unmixed and retained the gene order of the ancestral elements. We found that the number of species-specific fusions is significantly greater than expected under a uniform model of evolution across the phylogeny (see Methods). The scarcity of older fusions suggests that lineages with fusions have a reduced probability of persisting over time. Alternatively, fusions could revert via subsequent fission but we found no instances where reversion was a parsimonious explanation of observed chromosomes.

We explored whether all Merian elements were equally likely to be involved in fusions. For this analysis, only chromosomes resulting from a single fusion event between two elements were considered and

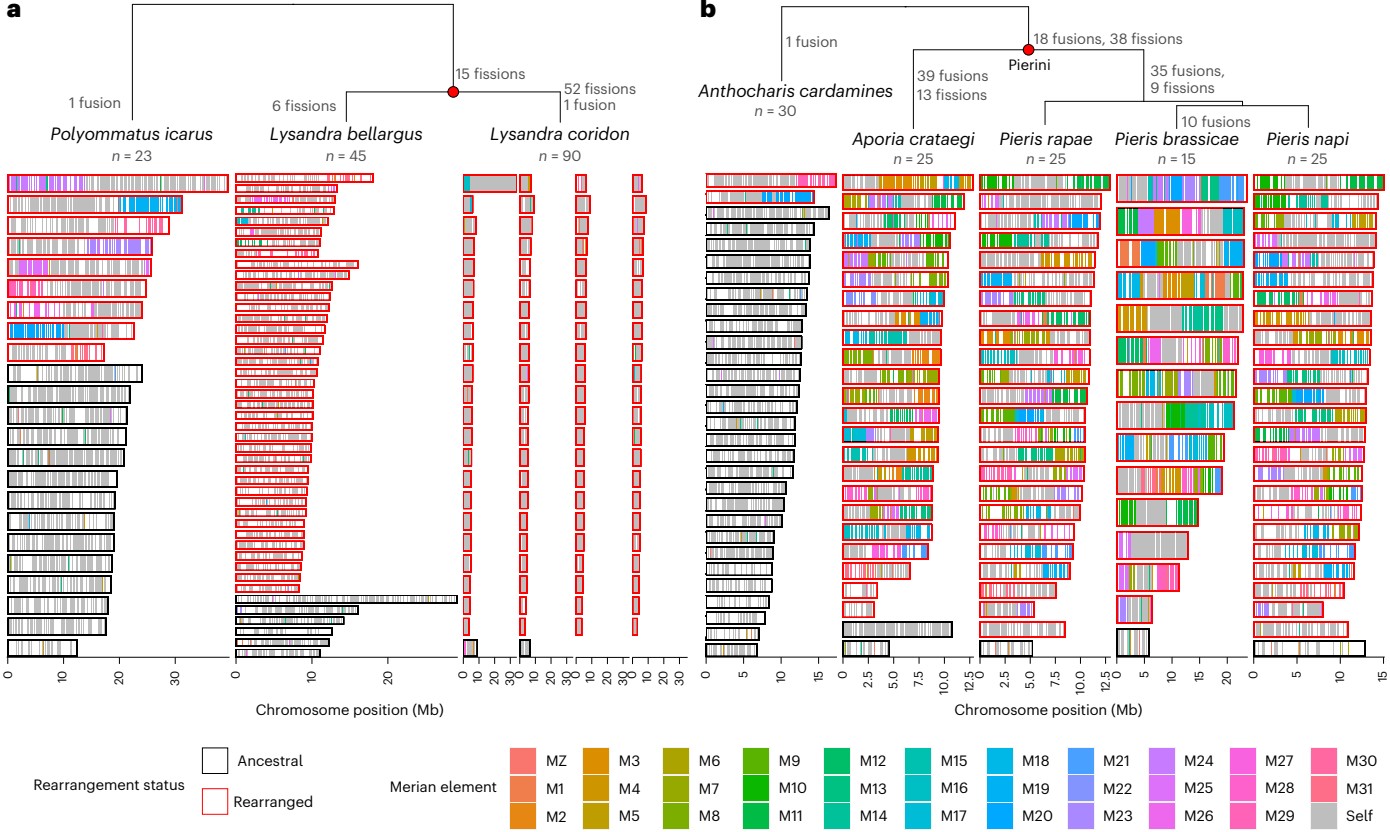

**Fig. 4 | Extensive chromosomal rearrangements in *Lysandra* and Pierini.**
**a**, Relationships of Lysandra species with reorganized genomes and sister species
*Polyommatus icarus* that has retained intact Merian elements with the exception
of seven fusions shared by all lycaenids. **b**, Relationships of the Pierini species
that have reorganized genomes and their sister species, *Anthocharis cardamines*,
which is not reorganized. In both panels, Merian elements are painted across the
chromosomes of each species. Each chromosome is represented by a rectangle
within which the position of each orthologues is painted grey if it belongs to the
most common Merian element for that chromosome or coloured if it belongs to
an alternative Merian element. Chromosomes that have undergone large-scale
rearrangements (fusions or fissions) are outlined in red. The full list of identified
rearrangements is available in Supplementary Table 7. Source data for this figure
can be found in Supplementary Table 10 and in the Zenodo repository[122].

the ancient fusion observed in all Ditrysia was considered as one unit.
We found that some Merian elements were more frequently involved
in fusion than others (Fig. 3b). The most common fusion pairings were
MZ + M31 and MZ + M24 (each with four independent occurrences).
Strikingly, MZ was involved in the highest number of fusion events (30
independent fusion events). We found that small autosomal elements
were involved in more fusion events than were larger ones (Spearman's
rank correlation, $\rho(29) = -0.62$, $P = 2 \times 10^{-4}$) (Fig. 3c and Extended Data
Fig. 3). A bias towards the involvement of smaller chromosomes in
fusion events has been found in *Bombyx mori* and *Heliconius mel-
pomene*[42]. Our analysis suggests that this holds across Lepidoptera and
is true for both autosome–autosome fusions and Z–autosome fusions.

## Extensive rearrangements in eight independent lineages

Against the backdrop of strong constraint on karyotype evolution,
14 species from 8 lineages had highly reorganized genomes (Fig. 1a
and Supplementary Table 7). We identified two distinct patterns, one
exemplified by *Lysandra*, where fission has been dominant (Fig. 4a)
and the other by tribe Pierini (Pieridae), where chromosomes have
undergone many nested fusion and fission events (Fig. 4b). Both pat-
terns have resulted in fragmentation of Merian elements. We found no
evidence of polyploidy in any lineage.

To investigate the dynamics of fission in Lysandra (Nymphalidae),
we reconstructed the events that gave rise to the genome structures
of *Lysandra coridon* and *Lysandra bellargus*. Seven pairwise fusions
generated a karyotype of n = 24 in the last common ancestor of family

Lycaenidae. Fifteen fissions then generated n = 39 in the last common
ancestor of Lysandra (Fig. 4a). Subsequently, *L. bellargus* underwent
six fissions generating n = 45 and *L. coridon* experienced at least one
fission event in 37 of the 39 chromosomes of the *Lysandra* last common
ancestor. The MZ element did not undergo fission in either species but
fused to a portion of M16 in *L. coridon*. An overwhelming majority of
the 90 chromosomes in *L. coridon* mapped to a single Merian element
and show conservation of gene order (Supplementary Fig. 9). The few
*L. coridon* chromosomes that contained segments from more than one
Merian element derive from the seven fused chromosomes present in
the common ancestor of Lycaenidae. A similar pattern of dominance
of fission was observed in *Tinea semifulvella*, which has undergone
15 fission events, resulting in a karyotype of n = 45 relative to *Tinea
trinotella* (n = 30) (Supplementary Fig. 10).

In Pierini (Pieridae), chromosomes are mosaics of segments of
Merian elements. We inferred parsimonious rearrangement histories
that explain the karyotypes of *Pieris napi, Pieris rapae, Pieris brassicae*
and *Aporia crataegi* (Fig. 4b). A set of fusions and fissions occurred in
the last common ancestor of Pierini and are thus absent in the outgroup
*Anthocharis cardamines*. Further fusions and fissions occurred inde-
pendently in the lineages leading to *A. crataegi* and to the three *Pieris*
species. *P. rapae* and *P. napi* share 25 orthologous, collinear chromo-
somes and thus have maintained the same karyotype as the last com-
mon ancestor of *Pieris* for ~30 million years[49]. In contrast, *P. brassicae*
underwent ten more fusions resulting in a reduced karyotype of n = 15.

Complex, nested rounds of fusion and fission have also shaped the
genomes of *Melinaea* (Nymphalidae). A series of fusions and fissions

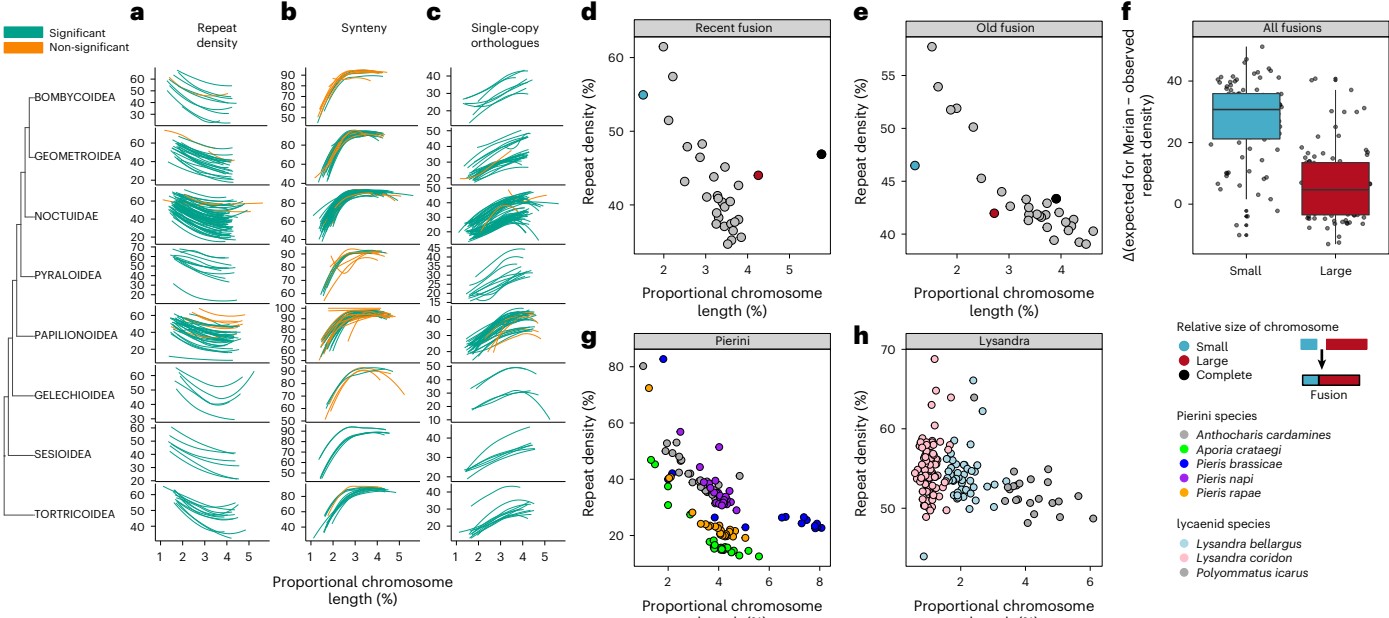

**Fig. 5 | The correlates of chromosome length and sequence features across Lepidoptera. a–c**, Proportional chromosome length against sequence features: repetitive element content (**a**); synteny, defined as the proportion of orthologues that are adjacent in both the reference species *Melitaea cinxia* and the given species (**b**); proportion of chromosomal gene content that is made up of orthologues that are single copy and present across Lepidoptera (**c**). Each line is coloured green if the correlation with proportional length was significant (Spearman's rank, *P* < 0.05) or orange if it was non-significant (Supplementary Table 8). Spearman's rank correlation coefficients (*R*) and *P* values were obtained by two-sided Spearman's correlation test. Only autosomes were included in the correlation analysis. Autosomes were filtered to only retain those that corresponded to intact Merian elements (that had not undergone fusion or fission). Only species with at least ten autosomes after filtering were analysed and only superfamilies represented by at least five species are shown.

**d,e**, Proportional chromosome length against repetitive element content for *Agrochola circellaris* (**d**), which has a recent fusion and for *Aphantopus hyperantus* (**e**), which has an older fusion. **f**, The difference between the average repeat density of a given Merian element and its current repeat density in the context of a fused chromosome is shown, where small Merian elements are M25, M29, M30 and M31. The box plots show the median (centre line) and the first and third quartiles (Q1 and Q3; box limits) and the whiskers extend to the last point within 1.5 times the interquartile range below and above Q1 and Q3 respectively. Observations that fall outside Q1 and Q3 are shown as outliers. *n* = 180 independent pairwise fusions examined. **g,h**, Proportional chromosome length against repetitive sequence content is shown for a set of Pierini species plus the sister species *Anthocharis cardamines* (**g**) and for species in genus *Lysandra* and the sister species *Polyommatus icarus* (**h**). Source data for this figure can be found in Supplementary Tables 10 and 8 and in the Source data.

occurred in the last common ancestor of *Melinaea*, with further independent fusions and fissions occurring in *Melinaea marsaeus* and *Melinaea menophilus* (Supplementary Fig. 11). Likewise, the genomes of *Brenthis ino* (Nymphalidae) and *Apeira syringaria* (Geometridae) reflect a history of many fusions and fissions, having undergone an estimated total of 33 and 38 events, respectively (Supplementary Figs. 12 and 13). *Leptidea sinapis* (Pieridae) has undergone 29 fusion and 26 fission events, resulting in *n* = 48 compared to its close relative, *Anthocharis cardamines*, which has *n* = 30 (Supplementary Fig. 14). Two closely related species in Geometridae, *Operophtera brumata* and *Philereme vetulata*, had highly reorganized genomes. We infer that three fissions occurred in their last common ancestor. *O. brumata* experienced a further 11 fissions and 30 fusions. In contrast, one fusion and 35 fissions occurred in *P. vetulata* (Supplementary Fig. 15). Notably, in all highly reorganized lineages, MZ has remained intact with no fissions and in all lineages, except *P. vetulata*, it has fused to one or more autosomal Merian elements.

## Understanding biases in chromosomal fusions in Lepidoptera

Small and sex-linked Merian elements are more frequently involved in fusion events. This leads to the question of whether there are compositional differences that vary with chromosome length. We observed a negative correlation between GC content and proportional chromosome length in 84% (163 of 193) of analysed species (Spearman's rank, *P* < 0.05) (Supplementary Table 8 and Extended Data Fig. 4a) with small chromosomes having high GC content. GC content has several drivers,

including contributions from repetitive elements but GC3 (the GC content of the third bases of potentially degenerate codons) is independent of many of these. Only half (48%; 93 of 184) of the species analysed had higher GC3 values in smaller chromosomes (Supplementary Table 8 and Extended Data Fig. 4b) suggesting that some variation in GC is driven by the density of features such as repeats. Consistent with this, smaller chromosomes have a higher repeat density than larger chromosomes (Fig. 5a). Negative correlation between chromosome length and repeat density was observed in 93% (180 of 193) of assayed species (Spearman's rank, *P* < 0.05), ranging in strength from −0.41 (*Notocelia uddmanniana*) to −0.98 (*Biston betularia*) (Supplementary Table 8). High repeat density in smaller chromosomes was not associated with specific repeat types. All major repeat families were enriched in shorter chromosomes, albeit some families more so than others (Extended Data Fig. 5). In contrast to GC content and repeat density, we observed no consistent correlation between coding density and chromosome size (negative correlation in 0.5% (1 of 184) and positive correlation in 18% (33 of 184) of species; Spearman's rank, *P* < 0.05) (Extended Data Fig. 4c), reflecting previous conflicting trends observed in several Nymphalid species[50,51].

While gene order synteny is highly conserved in Lepidoptera, smaller chromosomes were generally less syntenic than longer chromosomes (Fig. 5b). A significant positive correlation (Spearman's rank, *P* < 0.05) was observed in 68% of species (132 of 193) with correlation strength ranging from 0.82 (*Limenitis camilla*) to 0.37 (*Chrysoteuchia culmella*) (Spearman's, *P* < 0.05) (Supplementary Table 8). We explored whether the types of genes on small chromosomes were different from

those on larger chromosomes. Smaller chromosomes were depleted in single-copy orthologues relative to larger chromosomes in 95% (174 of 184) of all analysed species (Spearman's rank, *P* < 0.05) (Fig. 5c).

In several of these analyses, the Z chromosome was an outlier given its relative length (Supplementary Table 9). Unfused MZ chromosomes had low average GC and GC3 content, in line with GC decreasing with chromosome length (Extended Data Fig. 6a,b). However, the average repeat content for MZ chromosomes was higher than expected on the basis of chromosome length alone (Extended Data Fig. 6c). Although the level of coding density on MZ chromosomes fell within the range exhibited by autosomes (Extended Data Fig. 6d), they had a much lower level of synteny than expected on the basis of chromosome length (Extended Data Fig. 6e). MZ chromosomes were also relatively depleted in single-copy, conserved genes (Extended Data Fig. 6f). Together, these patterns indicate that other evolutionary forces, in addition to the chromosome length, have shaped the content of the Z chromosome.

## Consequences of fusions

The composition of Merian elements might be an intrinsic part of their functional biology rather than driven by their relative sizes. Intrinsic function would maintain Merian element-specific feature landscapes in fused chromosomes, while length-related drivers would result in amelioration through time. For phylogenetically recent fusions, we observed that the constituent Merian elements had a repeat density similar to that of their ancestral, unfused homologues. For example, in the species-specific M30 + M5 fusion in *Agrochola circellaris* (Noctuidae) we found a repeat-rich M30 segment and a larger and relatively repeat-poor M5 segment. The repeat densities of these segments were in line with expectations from the ancestral, unfused sizes (Fig. 5d). As noted above, chromosomes have higher repeat densities at their ends and, in recent fusion chromosomes, a shoulder of higher repeat density in the area of the fusion (probably a relic from the contributing parts) was evident (Extended Data Fig. 7a–c). *Aphantopus hyperantus* (Nymphalidae) had a phylogenetically older M29 + M14 fusion that was shared by members of subfamily Satyrinae. While the M29- and M14-derived domains of the fused chromosome were still distinct in syntenic gene content, they both had repeat densities consistent with an expectation derived from the fused chromosome length (Fig. 5e). There was no central shoulder of increased repeat density (Extended Data Fig. 8). In all simple fusions involving one of the four smallest Merian elements, the smaller Merian element tended to have experienced a greater shift in repeat density relative to its unfused ancestor (paired *t*-test, *P* < 0.01) (Fig. 5f). Thus, the repeat landscape of fused chromosomes evolves over time to reflect that expected of larger chromosomes. Patterns of features on chromosomes are therefore largely driven by the relative chromosome length, not the identities of the genes carried.

Average chromosome length will be smaller in species with more chromosomes and thus would be expected to accumulate a higher density of repeats. The small, highly reorganized chromosomes of Pierids were indeed repeat-rich relative to the chromosomes of *Anthocharis cardamines* (Fig. 5g) and the small chromosomes resulting from rampant fission in groups such as Lysandra were also repeat-rich (Fig. 5h). Despite the lack of correlation between chromosome number and genome size across all species, repeat accumulation in species with many, smaller chromosomes was associated with an increase in genome size in *Lysandra* species (Supplementary Fig. 16), *L. sinapis* (Supplementary Fig. 17), *P. vetulata* (Supplementary Fig. 18) and *T. semifulvella* (Supplementary Fig. 19). Symmetrically, reduction in chromosome numbers was associated with reduced genome size in Pierini (Supplementary Fig. 20), *A. syringaria* (Supplementary Fig. 21) and *B. ino* (Supplementary Fig. 22) but not in *O. brumata* (Supplementary Fig. 18) and *Melinaea* species (Supplementary Fig. 23). It may be that the many fusions that reduce chromosome number in these last species were recent and insufficient time has passed for repeat content to decrease.

## Discussion

The ongoing revolution in sequencing is enabling major projects such as the Darwin Tree of Life to produce large numbers of chromosomally complete genomes across eukaryotic diversity[36,52]. These rich data permit comprehensive, large-scale, taxon-wide analysis of features and processes[53]. Using over 200 chromosomally complete genomes, we mapped the evolutionary dynamics of chromosome maintenance, fusion and fission in a holocentric group, the Lepidoptera. We found that the chromosomes of extant species are derived from 32 ALGs or Merian elements. Except for an ancient Ditrysian fusion, Merian elements have remained intact in most species. Our findings complement previous work that demonstrated strong conservation of macrosynteny in Lepidoptera[41–43] by defining their precise orthologue content. These elements have consistent differences in genomic features and carry distinct sets of conserved genes that retain a syntenic order. Merian elements provide a unifying system to explore genomic stasis and change in Lepidoptera, similar to Müller elements of *Drosophila* and Nigon elements of rhabditid nematodes[3,4,54,55].

Across Lepidoptera, we find that fusions are rare and fissions rarer still. Surprisingly, we found relatively few fusions on deeper branches of the phylogeny, consistent with lineages possessing fusions being less likely to persist. Alternative explanations, such as a general increase in the rate of fixation of fusions in recent time or frequent reversion by exact fission seem unlikely. We note that this analysis is based on a fraction of Lepidopteran diversity and requires deeper investigation with denser species sampling. We also found that Lepidopteran chromosomes arising from fusions retain syntenic domains that reflect the original elements. Remarkably, this includes the M17 + M20 fusion, which occurred ~200 million years ago. In contrast, holocentric chromosomes in nematodes have a high rate of intrachromosomal rearrangement that leads to rapid mixing of genes from Nigon elements in fused chromosomes[3,4]. We find that smaller Merian elements are more often involved in fusion events than are larger autosomal elements. The distinct relative sizes of Merian elements also mean that they evolve differently. In Lepidoptera, each bivalent typically undergoes one meiotic recombination in males[56,57], meaning that smaller Merian elements experience higher per base recombination rates than longer elements. In addition to reducing linkage disequilibrium and enhancing the efficacy of selection, recombination is mutagenic[58], meaning smaller elements will experience higher mutational pressures. The stability of Merian element size across Lepidoptera means that these differences will have had a long-term impact on the evolutionary trajectories of the genes and genetic systems each element carries and elements that fuse or split will experience a step-change in evolutionary rates. Consistent with this, fused Nymphalidae chromosomes have decreased nucleotide diversity compared to their unfused homologues in sister species[39] and raised barriers to introgression[51,59,60].

Small Merian elements show some similarities to the monocentric, GC-rich microchromosomes of vertebrates[61]. Interestingly, comparative analyses indicate that the ancestral vertebrate possessed a set of small gene-rich chromosomes. Subsequently, subsets of microchromosomes progressively fused, resulting in macrochromosomes. Therefore, our finding of the involvement of small chromosomes in genome reorganization across Lepidoptera shows some similarity to vertebrate chromosome evolution. However, unlike small Lepidopteran chromosomes, vertebrate microchromosomes are repeat-poor and gene-rich.

In our dataset, MZ was usually the largest chromosome and had sequence patterns that diverged from expectations derived from the longer autosomes, including repeat and gene content, and degree of synteny. Because of achiasmatic oogenesis, 67% of the population of MZ elements undergo crossovers each generation, in contrast to only 50% of the population of autosomal elements. The elevated recombination rate of the Z and haploid exposure in females probably explain these patterns[62]. Z–autosome fusions have previously been described in many lepidopteran species[63–65]. We corroborate these studies by

demonstrating that MZ has a higher rate of fusions than any autosomal element across Lepidoptera. Sex chromosome–autosome fusions are also overrepresented in rhabditine nematodes[4], flies[55], vertebrates[66] and plants[67]. Possible drivers include female meiotic drive[68], sexually antagonistic selection[69] and deleterious mutation sheltering[70–72]. The set of 30 independent MZ–autosome fusions described here presents a valuable dataset for dissection of the drivers of the rate of molecular evolution in sex chromosomes and, for fusions, illumination of the forces that shape autosomes. The resistance of MZ to fission in species where fission is dominant also requires deeper exploration.

Why have Merian elements remained largely stable in gene content and order through ~250 million years[73,74] of lepidopteran evolution? Species with holocentric chromosomes are theoretically more permissive to karyotypic change. This is reflected in some holocentric groups, such as *Carex* sedges, where karyotype evolution is rapid[75,76] but clear differences are not seen in monocentric versus holocentric insects[27]. One potential constraint on the fixation of rearrangements is the ability to undergo meiosis. Individuals heterozygous for rearrangements can be sterile due to unbalanced segregation leading to heterozygote disadvantage (underdominance)[77–79]. Structural heterozygosity impacts reproductive fitness in holocentric *Caenorhabditis elegans* nematodes and *Carex*[80,81]. Homologue pairing and kinetochore activity have been suggested to constrain karyotype evolution[82–84]. In *C. elegans*, homologue pairing is restricted to discrete regions enriched for short-sequence motifs while kinetochores assemble across regions of low transcriptional activity. While pairing centres have not been found in Lepidoptera, kinetochore assembly in *B. mori* is non-sequence-specific and occurs in regions with low transcriptional activity[85]. Understanding lepidopteran kinetochore and pairing centre biology will illuminate the roles of these basic systems in constraining or promoting chromosome number evolution.

Merian elements may be maintained to facilitate *cis*-regulation between genes. This has been suggested in vertebrates where the gene-rich microchromosomes experience a lower interchromosomal rearrangement rate than their larger counterparts[86–88]. It has been suggested that the syntenic blocks of genes resulting from fusion and fission in *Pieris* represent gene sets with related functions and these networks present a constraint[47]. Consistent with this, fusions disrupt patterns of chromosomal contacts in mouse germ cells[89] and rearrangement hotspots exist at the boundaries of topologically associated domains in mammalian chromosomes[88]. However, topologically associated domains are usually much shorter than individual chromosomes and so are unlikely to offer a complete explanation of Merian element conservation.

Chromosome evolution in Lepidoptera is not homogenous. Against a background of stasis, we find eight lineages that have experienced major change. We classify these lineages into autosomal fission-only, with extensive fission of autosomal elements resulting in many small autosomes and a large, intact MZ, or fission–fusion, with many fission and fusion events. In all lineages, MZ was insulated from fission. In the fission–fusion lineages, we also identified re-establishment of karyotype stability, albeit at chromosome numbers other than $n = 31$–32. For example, after fission and fusions, *Pieris* species restabilized at $n = 25$, with most *Pieris* species possessing this karyotype[26]. The three processes which generate lepidopteran chromosomal complements, karyotype-stabilizing constraint and karyotype-diversifying fission and fusion, can be separately modified in different lineages. For example, the mechanisms preventing fission were derepressed in *Lysandra* and fission and fusion were derepressed but fusion was more recently dominant in *P. brassicae*. Elevated rates of fixation of rearrangements may be a product of neutral processes such as genetic drift of mildly deleterious and/or underdominant changes during sustained periods of low effective population size[90]. Alternatively, functional differences in core chromosome biology could drive change. In parrots (Aves; Psittaciformes), frequent rearrangements have been linked to

the loss of genes involved in the repair of double-strand breaks and genome stability maintenance[91]. The existence of lepidopteran lineages where fission and fusion rates have been individually modified will permit detailed investigation of their mechanistic bases. We note that several species with highly reorganized genomes display variable karyotypes between populations[92,93], where mating between individuals with highly divergent karyotypes can produce fertile offspring, suggesting that meiosis in some lepidopterans can tolerate heterozygosity for many rearrangements[22,94,95]. However, the persistence of hybrid zones between populations with different karyotypes indicates a fitness cost in hybrids[92]. Transposable elements are suggested to facilitate high rates of chromosome fusion[42,96,97] by promoting deletion, translocation and inversion[98]. The smaller lepidopteran autosomes, which are more frequently involved in fusions, do have higher repeat content but MZ, which has relatively low repeat density and fuses frequently, does not. The evidence of repeat involvement in lepidopteran fusions is equivocal, as an enrichment of LINEs at fusion boundaries observed in *L. sinapis*[97] may be a relic of recent chromosomal fusion and analysis of the *P. napi* genome found no enrichment of repeats at fusion boundaries and no repeat class was expanded compared to other species[47].

While the impacts of karyotype on evolutionary trajectories may be indirect, their effects can be profound. All other things being equal, change in karyotype between species is unlikely to be neutral. Fundamentally, change probably promotes speciation[38]. However, the pattern of overall stasis indicates that lineages with highly variant karyotypes may be at a macroevolutionary disadvantage despite any short-term speciation advantage. Interestingly, karyotype analyses suggest that species with high rates of chromosomal change have both the highest speciation rates and the highest species turnover reflecting higher extinction rates[38], potentially consistent with unstable diversification with extinction over time. We highlight that higher chromosome counts mean more recombination and thus potentially faster evolutionary rates (or more effective selection) overall. This effect will be particularly marked for genes on elements directly involved in fusions and fissions and genome-wide in extensively rearranged species. Dense genomic sampling of closely related species that differ in rearrangements or, better still, individuals heterozygous for rearrangements, will provide a greater understanding of the immediate consequences of interchromosomal rearrangements on three-dimensional genome structure, recombination rate and the role of specific sequence features. Understanding the drivers and constraints of chromosome change expands our understanding of genome evolution and the role of chromosomal change in the evolution of diversity across the tree of life.

## Methods

### Chromosomal genome assemblies, annotations and transposable elements identification

We downloaded all representative chromosome-level reference genomes for Lepidoptera and Trichoptera that were available on INSDC on 27 June 2022. Of these 212 lepidopteran genomes and 4 trichopteran genomes, 191 were generated by the Darwin Tree of Life Project[36]. Accession numbers and references for all genomes are given in Supplementary Table 1. For species generated by the Darwin Tree of Life project that do not have a reference, the methods were the same as for ref. 99. We used the primary assembly for all analyses. The speciose Noctuoidea (71 species) and the intensely studied Papilionoidea (51 species) contribute most to the genomes.

Gene annotations were generated by Ensembl[100] (http://rapid.ensembl.org) for 201 species (Supplementary Table 3). Species that had publicly available RNA sequencing (RNA-seq) data were annotated using Genebuild, which makes use of both RNA-seq and protein homology evidence. For species that did not have transcriptomic data, the genomes were annotated using BRAKER2 (ref. 101) using protein homology information as evidence. Protein data consisted of OrthoDB (v.11) data[102] for Lepidoptera combined with all lepidopteran proteins

with protein evidence levels 1 or 2 from UniProt[103] (where level 1 or 2 represent evidence from either proteomic or transcriptomic data). Details of each annotation are provided in Supplementary Table 3. The gene sets contained between 9,267 (*Tinea trinotella*) and 23,879 (*Miltochrista miniata*) protein-coding genes and between 15,416 (*Erynnis tages*) and 41,125 (*Dendrolimus punctatus*) transcripts. Transposable elements (TE) were identified using the Earl Grey TE annotation pipeline (v.1.2)[104,105] on each genome as described in ref. 106, with the Arthropoda library from Dfam release 3.5 (refs. 107,108).

Two genomes were excluded from further analysis due to quality issues. The first, *Zerene cesonia* (GCA 012273895.2), contained 246 unlocalized scaffolds that contained 351 BUSCOs. The high number of BUSCOs in these scaffolds means that erroneous rearrangement events would be inferred if this genome were to be included. In the second, *Cnaphalocrocis medinalis* (GCA 014851415.1), most genes belonging to the M30 Merian element were present on unlocalized scaffolds. We identified two more genomes that contained minor misassembly issues that we were able to address before downstream analysis (Supplementary Text). In *Dendrolimus kikuchii* (GCA 019925095.1), we found two scaffolds with a high proportion of duplicated BUSCOs (most of which corresponded to the M30 Merian element), indicating that they represented haplotypic duplication. When we removed these scaffolds from the assembly, we successfully recovered a fusion between M30 and MZ that would have otherwise been missed. In *Spodoptera frugiperda* (GCA 011064685.2), we removed an unlocalized scaffold that contained 22 BUSCOs before downstream analyses to avoid inferring a fission event in this species due to assembly issues.

## Phylogenetic tree reconstruction

We used BUSCO (v.5.4.3) (using the metaeuk mode and the lepidoptera odb10 dataset)[37] to identify single-copy orthologues in each genome. We used busco2fasta.py (available at https://github.com/lstevens17/busco2fasta) to identify 5,046 BUSCO genes that were single copy and present in at least 90% of the genomes. We aligned the protein sequences of these BUSCOs using MAFFT (v.7.475)[109] and trimmed alignments using trimal (v.1.4)[110] with parameters -gt 0.8, -st 0.001, -resoverlap 0.75, -seqoverlap 80. A total of 4,947 alignments passed the alignment thresholds. We concatenated the trimmed alignments to form a supermatrix using catfasta2phyml (available at https://github.com/nylander/catfasta2phyml). We provided this supermatrix to IQ-TREE (v.2.03)[111] to infer the species tree under the LG substitution model[112] with gamma-distributed rate variation among sites and 1,000 ultrafast bootstrap replicates[113]. The tree was rooted on the node separating Trichoptera and Lepidoptera and visualized alongside genome size and chromosome number information using ggtree (v.3.0.2)[114,115].

To test for a correlation between genome size and chromosome number, we used a phylogenetic linear model using the R package phylolm (v.2.6.2)[116] with genome size as the response variable and chromosome number as a fixed factor. To account for shared ancestry between species, the phylogenetic tree described above was included. The most appropriate model for the error terms was identified as Ornstein–Uhlenbeck (OU) by fitting all implemented models that allow for measurement error and then selecting the best-fitting model via the AIC values.

## Defining and visualizing Merian elements

We inferred the ancestral lepidopteran linkage groups using syngraph (available at https://github.com/A-J-F-Mackintosh/syngraph)[40] (using a threshold of five orthologues and using the mode that infers fusions and fission events) using the BUSCO-derived single-copy orthologues and the phylogeny derived from all 210 chromosomal lepidopteran genomes and 4 chromosomal trichopteran genomes. As described in ref. 40, syngraph uses parsimony to infer the arrangement of orthologues in the last common ancestor of species triplets. Syngraph works from the tips towards the root to infer ALGs (and fusion and

fission events, discussed below) at each internal node in the tree. We used the ALGs inferred by syngraph in the last common ancestor of all Lepidoptera in our analysis, which we termed Merian elements. We named Merian elements in ascending order on the basis of the number of orthologues contained (M1–M31). The group of orthologues that represented the ancestral Z chromosome were named MZ. We 'painted' the chromosomes of each extant species to show the distribution of these Merian elements using custom scripts (available at https://github.com/charlottewright/lep_busco_painter). Merian elements also can be painted onto a given genome via the interactive website https://charlottejwright.shinyapps.io/busco_painter/. We also visualized synteny between pairs of species using Oxford plots generated using custom scripts (available at https://github.com/charlottewright/Chromosome_evolution_Lepidoptera_MS).

To assess the extent to which our orthologue assignments to Merian elements is dependent upon species sampling, we performed a bootstrap analysis. We performed 100 iterations of ancestral unit inference using syngraph, each time with a different random set of 110 (50%) of Lepidopteran species. As the ancient fusion of M17 and M20 is only apparent when including the Trichoptera representatives as outgroups and *M. aruncella*, we kept these species in each iteration. We recovered 32 linkage groups in all 100 iterations. There was not a single conflicting orthologue assignment in any of the 100 iterations (that is, no orthologue was assigned to a different Merian element). The only variation between iterations was the number and identity of orthologues that were unassigned. On average, each Merian-defining orthologue was unassigned in 12% of iterations, which probably arises from stochastic absences or duplication in the sampled species.

We also verified the accuracy of our orthologue assignments by performing ancestral genome reconstruction using AGORA (v.3.1)[45] which, in addition to inferring linkage, also reconstructs gene order. The input for AGORA was prepared by running 'convert_buscos.py' on the set of 214 BUSCO tables. The resulting orthologue groups and the species tree were then used to run 'agora-basic.py'. All 4,112 Merian-defining orthologues were in the reconstruction of the last common ancestor of Lepidoptera from AGORA. Of these, AGORA placed 3,092 into 683 contiguous ancestral regions (CARs). All CARs contained orthologues mapping to a single Merian element, with the exception of a single CAR (CAR_68) which contained nine orthologues belonging to M4 and one conflicting orthologue which corresponded to M1 (Supplementary Fig. 6). The results from AGORA therefore correspond extremely closely to Merian elements inferred from syngraph, with 99.97% agreement (3,091 of 3,092). We opted not to use the AGORA output because it was highly fragmented (Supplementary Fig. 7) due to the fact that AGORA requires gene order conservation and many small gene order differences exist between *M. aruncella* and *T. trinotella*.

## Comparison of Merian elements to bilaterian linkage groups

To assess the extent to which Merian elements are conserved beyond Lepidoptera, we compared Merian elements to the ALGs of Bilateria[7]. To do so, we first downloaded the gene annotation for *Tribolium castaneum* from Ensembl Metazoa (release 75) and filtered the protein annotation file using AGAT (v.1.0.0)[117] to retain only the longest isoform per gene. We then inferred 1:1 single-copy orthologues with the isoform-filtered protein file of *Melitaea cinxia* (Supplementary Table 3) by running OrthoFinder (v.2.5.4)[118] on the two sets of proteins. The single-copy orthologues were filtered to only retain those which had been assigned to a BLG in ref. 7. To compare to Merian elements, we ran BUSCOs (v.5.4.3) (using the lepidoptera odb10 dataset) on the protein set of *M. cinxia* and filtered the output to only retain orthologues which are assigned to both Merian elements and BLGs. This resulted in a set of 916 orthologues. To assess whether Merian elements are associated with BLGs, we assessed the variation in distribution of orthologues from given Merian elements across the set of 24 BLGs. To construct a null distribution of orthologue assignment, we performed 100,000

simulations where the variance was calculated from a random distribution of the orthologues of a given Merian element across BLGs, weighted by the number of orthologues per BLG. We then compared this distribution of variance to the observed variance in the distribution of the orthologues of each Merian element across the 24 BLGs. We considered observed levels of variance above a 99.99% percentile as significantly higher than expected under a null distribution of random assignment.

### Inferring fusion and fission events

We inferred simple fusion and fission events (defined as those that involve complete Merian elements and did not appear to be nested) using two complementary approaches: syngraph[40] and lep_fusion_fission_finder (LFFF) (available at https://github.com/charlottewright/lep_fusion_fission_finder). As discussed above, syngraph infers ALGs at each internal node in the tree along with any fusion and fission events that occurred at each branch. In contrast, LFFF uses a set of ALGs (in this case, the Merian elements inferred by syngraph) to identify fused or split chromosomes in extant species only. To do this, LFFF identifies the most common Merian element in non-overlapping windows of a given size. Fused chromosomes are identified as those containing windows assigned to two or more Merian elements (and the position along the chromosome where Merian-element identity switches is recorded as the fusion position). Split chromosomes are identified as those in which a Merian is assigned to two or more chromosomes. Fusion and fission events are then inferred by mapping these fused and split chromosomes onto the phylogeny. We identified the optimal number of orthologues as a threshold in both syngraph and LFFF by manually assessing the inferred events. At low thresholds (<17), small rearrangement events or unlocalized scaffolds are often identified as fused chromosomes or split chromosomes. However, at higher thresholds (>17), small or split Merian elements are often erroneously excluded. We identified the optimal threshold at 17 for both syngraph and LFFF. Using this threshold, we obtained nearly identical results with both approaches, with the only differences being due to how fusions involving more than two Merian elements are denoted (Supplementary Tables 5 and 6). The genomes of species that had one or more examples in which orthologues belonging to a single Merian element were present along more than one chromosome and in which such chromosomes are not the product of simple fission events, were classified as highly rearranged species with complex rearrangements and so were analysed separately. Similarly, species with genomes resulting from many fission events, leading to at least one chromosome with fewer Merian-defining orthologues than our threshold (<17), were classified as highly rearranged and so analysed separately. To analyse these species, syngraph was run on the complete set of 210 lepidopterans and 4 trichopterans using a lower, more sensitive threshold of five orthologues.

We tested whether an excess of fusions was inferred to be species-specific, that is occurred along external branches, by simulating a null distribution of fusion events over the lepidopteran phylogeny using a custom script (available at https://github.com/charlottewright/Chromosome_evolution_Lepidoptera_MS). To do this, 100,000 simulations were performed where the branch lengths were recorded over the phylogeny and whether the branch was external or internal. Then, a random sample of 183 fusions were weighted by branch lengths, with the assumption that fusions happen uniformly across the tree. The number of the 183 fusions that were on external branches versus internal branches was then recorded and compared to the observed number of events on external branches. We considered a number of fusion events on external branches above a 99.99% percentile as significantly higher than expected under a uniform distribution of fusions across the phylogeny.

The strength of rank-based correlation between the average proportional chromosome length of each Merian element and the frequency of fusion events was calculated using Spearman's rank implemented in the R package stats (v.4.1.0), with a $P < 0.05$ cutoff to assess significance[119].

### Describing feature distributions across chromosomes

We calculated the distribution of sequence features (GC, repeat density and coding density) along each chromosome 100 kb windows. GC content per 100 kb was calculated using fasta_windows (v.0.2.4) (https://github.com/tolkit/fasta_windows). For other features, a BED file specifying the start and end of each 100 kb window was generated for each genome with BEDtools (v.2.30.0)[120]. Repeat density was calculated using BEDtools coverage and the repeat annotation file produced by Earl Grey. To calculate coding density, we filtered the GFF3 files using AGAT (v.1.0.0)[117] to retain only the longest transcript per gene. As a quality check, we excluded CDS sequences that were not divisible by three using a custom Python script (available at https://github.com/charlottewright/genomics_tools). The resulting filtered GFF3 files were used with BEDtools coverage to calculate CDS density in 100 kb windows.

We also calculated the density of each feature by splitting each chromosome into 100 windows. First, a BED file specifying the position of each window along chromosomes was made using BEDtools makewindows with the fasta index file generated from samtools index (v.1.7)[121]. Repeat density was then calculated per window using BEDtools coverage (v.2.30.0)[120]. GC per window was calculated from the output from running fasta_windows (v.0.2.4) on 100 kb windows, using a custom Python script (https://github.com/charlottewright/Chromosome_evolution_Lepidoptera_MS) and the BED file containing the positions of each window.

### Describing feature distributions between chromosomes

We calculated the average density of various features (GC, GC3, repeat density, coding density, synteny and proportion of single-copy orthologues) in each chromosome (Supplementary Table 10).

The average GC content of each chromosome was calculated using fasta_windows (v.0.2.4) (https://github.com/tolkit/fasta_windows). To calculate the average GC3 value per chromosome, the GC3 value for each coding sequence was calculated using gff-stats (https://github.com/charlottewright/gff-stats/) and these values were used to calculate the average per chromosome using a custom Python script (available at https://github.com/charlottewright/genomics_tools/). Average repeat density per chromosome was calculated using BEDtools (v.2.30.0)[120].

We calculated the degree of synteny, defined as conserved gene order, per chromosome using a custom Python script (available at https://github.com/charlottewright/genomics_tools/). We calculated synteny as the proportion of adjacent gene pairs that have collinear orthologues in a corresponding species. We used the BUSCO genes defined previously and calculated synteny in each species relative to *Melitaea cinxia*.

The proportion of conserved single-copy orthologues relative to multicopy orthologues and species- or clade-specific genes was inferred from the annotated proteins obtained from Ensembl. We first filtered the GFF3 files for each species using AGAT to contain only the longest isoform per protein-coding gene. We filtered the corresponding protein files using fastaqual_select.pl (https://github.com/sujaikumar/assemblage). We then clustered all protein files into orthologous groups using OrthoFinder (v.2.5.4)[118]. By analysing these groups, we found that the annotation for one species, *Pieris napi*, was missing many orthologues present in most of the other annotations (Supplementary Fig. 1, 2). We therefore removed this annotation from the dataset and re-inferred orthologues with OrthoFinder. We identified 4,946 orthologous groups that were duplicated or missing in no more than 10% of species. We then classified each gene as either single copy, multicopy or clade-specific using a custom Python script (available at https://github.com/charlottewright/genomics_tools/). The classified genes were used to calculate the proportion of genes per chromosome that were classified as single copy versus non-single

copy using a custom Python script (available at https://github.com/charlottewright/genomics_tools/).

To compare the density of these features across lepidopteran chromosomes, we considered only those species which contained ten or more chromosomes that had not undergone a fusion or fission event (which left 193 species). Nine of these species did not have a publicly available gene annotation and so coding density, GC3 and proportion of single-copy orthologues could not be analysed. For each feature, the strength of the rank-based correlation between the feature value and proportional chromosome length (calculated as the chromosome length divided by the genome size) was calculated using Spearman's rank implemented in the R package stats (v.4.1.0), with $P < 0.05$ cutoff to assess significance.

## Repeat analysis within fusion chromosomes

To understand the effect of fusion on the repeat content of fused chromosomes, we chose fusions that involved M31, M30, M29 or M25 (which are the Merian elements with the lowest proportional length and were therefore expected to contain the highest repeat content). We expected the chromosomes involved in these fusion events to display the largest difference in repeat content before the fusion event. We created a BED file for each fused chromosome containing two windows, split at the fusion points that were defined by LFFF previously. The average repeat content for each window was calculated using BEDtools coverage. The difference between the repeat content of the larger-in-length Merian element and the smaller Merian element was statistically compared with a paired $t$-test as implemented in the R package stats (v.4.1), with $P < 0.05$ to cutoff to assess significance.

## Reporting summary

Further information on research design is available in the Nature Portfolio Reporting Summary linked to this article.

## Data availability

The reference genomes analysed in this study are available at https://www.ncbi.nlm.nih.gov/ and the accession numbers are given in Supplementary Table 1. Gene annotations are available at rapid.ensembl.org and are listed in Supplementary Table 3. The Arthropoda library from Dfam release 3.5 used to identify transposable elements is available at https://www.dfam.org/releases/Dfam_3.5. Large data files associated with this paper, including repeat annotations, repeat libraries and phylogenies are available at the Zenodo repository https://doi.org/10.5281/zenodo.7925505 (ref. 122). Other data supporting the findings presented in this paper are available in the Supplementary Tables as well as on GitHub (https://github.com/charlottewright/Chromosome_evolution_Lepidoptera_MS), which has been accessioned in Zenodo at https://doi.org/10.5281/zenodo.10373060 (ref. 123). Source data are provided with this paper.

## Code availability

The code associated with the analyses and figures can be found at https://github.com/charlottewright/Chromosome_evolution_Lepidoptera_MS, which has been accessioned in Zenodo at https://doi.org/10.5281/zenodo.10373060 (ref.123).

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

## Acknowledgements

M.B. and M.L. acknowledge funding support from Wellcome Trust grant no. 218328. C.J.W., L.S., M.L. and M.B. acknowledge funding support from the Wellcome Trust award 220540/Z/20/A 'Wellcome Sanger Institute Quinquennial Review 2021–2026'. A.M. acknowledges funding from the Natural Environment Research Council with an E4 PhD studentship (NE/S007407/1). We acknowledge the many people and groups which generated the publicly available chromosomal genomes that were used in this work, especially the Darwin Tree of Life project coordinated by the Wellcome Sanger Institute[36]. We thank M. Brown for advice on statistics and helpful discussions. We thank M. Muffato for help with running software at scale and are grateful to Tree of Life colleagues for reading and commenting on an earlier draft of this paper.

## Author contributions

C.J.W. and M.B. conceptualized the project. C.J.W., L.S. and A.M. developed the methodology. C.J.W., L.S. and A.M. conducted the investigation. C.J.W. and L.S. performed visualization. M.B. and M.L. acquired funding and administered the project. M.B., M.L. and L.S. supervised the project. C.J.W. wrote the original draft. C.J.W., L.S., A.M., M.L. and M.B. reviewed and edited the paper.

## Competing interests

The authors declare no competing interests.

## Additional information

**Extended data** is available for this paper at https://doi.org/10.1038/s41559-024-02329-4.

**Correspondence and requests for materials** should be addressed to Charlotte J. Wright or Mark Blaxter.

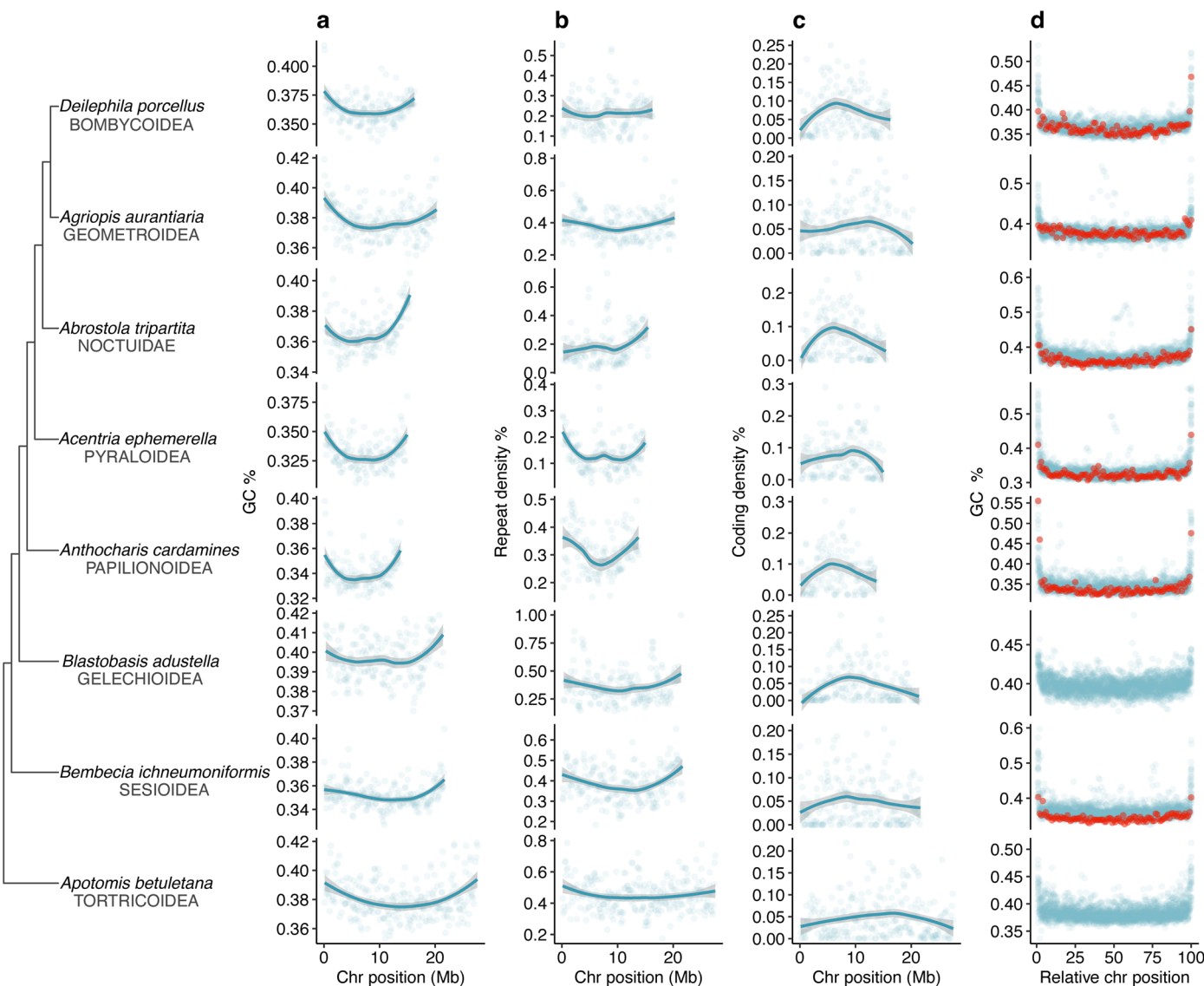

**Extended Data Fig. 1 | Sequence patterns in Lepidopteran chromosomes.**
**a**–**c**, Sequence patterns for one representative per superfamily in 100 kb non-overlapping windows; GC content in M1 (**a**), repeat density, considering all repetitive elements, in M1 (**b**) coding sequence density in M1 (**c**). d, GC content per chromosome in 100 windows per chromosome. Only chromosomes that have remained intact relative to Merian elements (that have not undergone fusion or fission) were included. Z chromosome values are coloured in red. Lines represent LOESS smoothing functions fitted to the data.

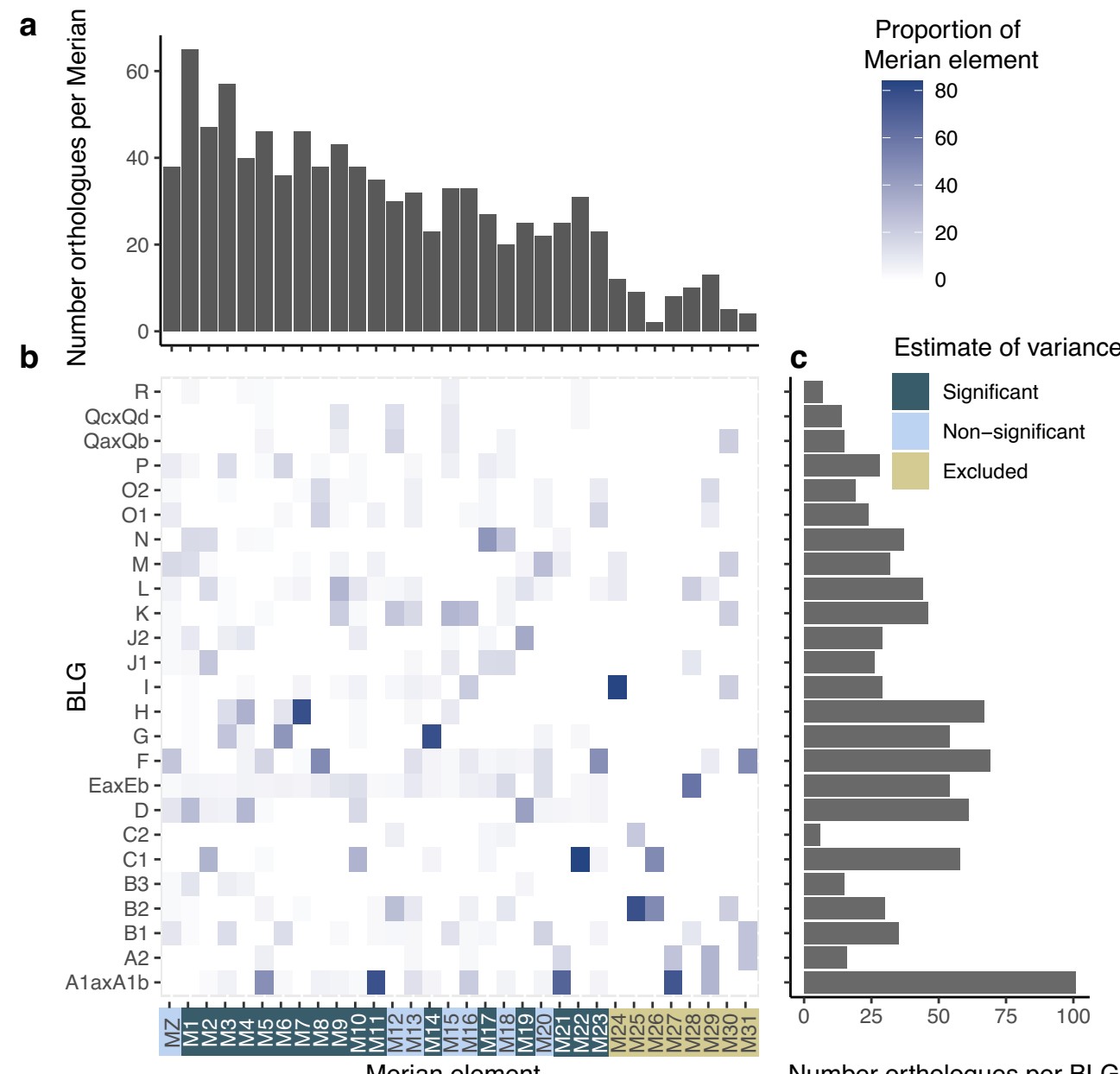

**Extended Data Fig. 2 | Conservation of Merian elements relative to bilaterian linkage groups. a**, Total number of orthologues per Merian element that could be assigned to a bilaterian linkage group (BLG). **b**, Heatmap of the proportion of orthologues of each Merian element found on each BLG. The labels of the Merian elements are coloured according to whether the Merian element showed a higher variation in the distribution of orthologues on BLGs than expected compared to a null model of random orthologue assignment weighted by the number of orthologues per BLG, indicating an uneven contribution of BLGs. Merian elements with less than 15 orthologues assigned to a BLG could not be assessed because they had too few data points to model a null distribution. **c**, Total number of orthologues per bilaterian linkage group (BLG) that could be assigned to Merian elements.

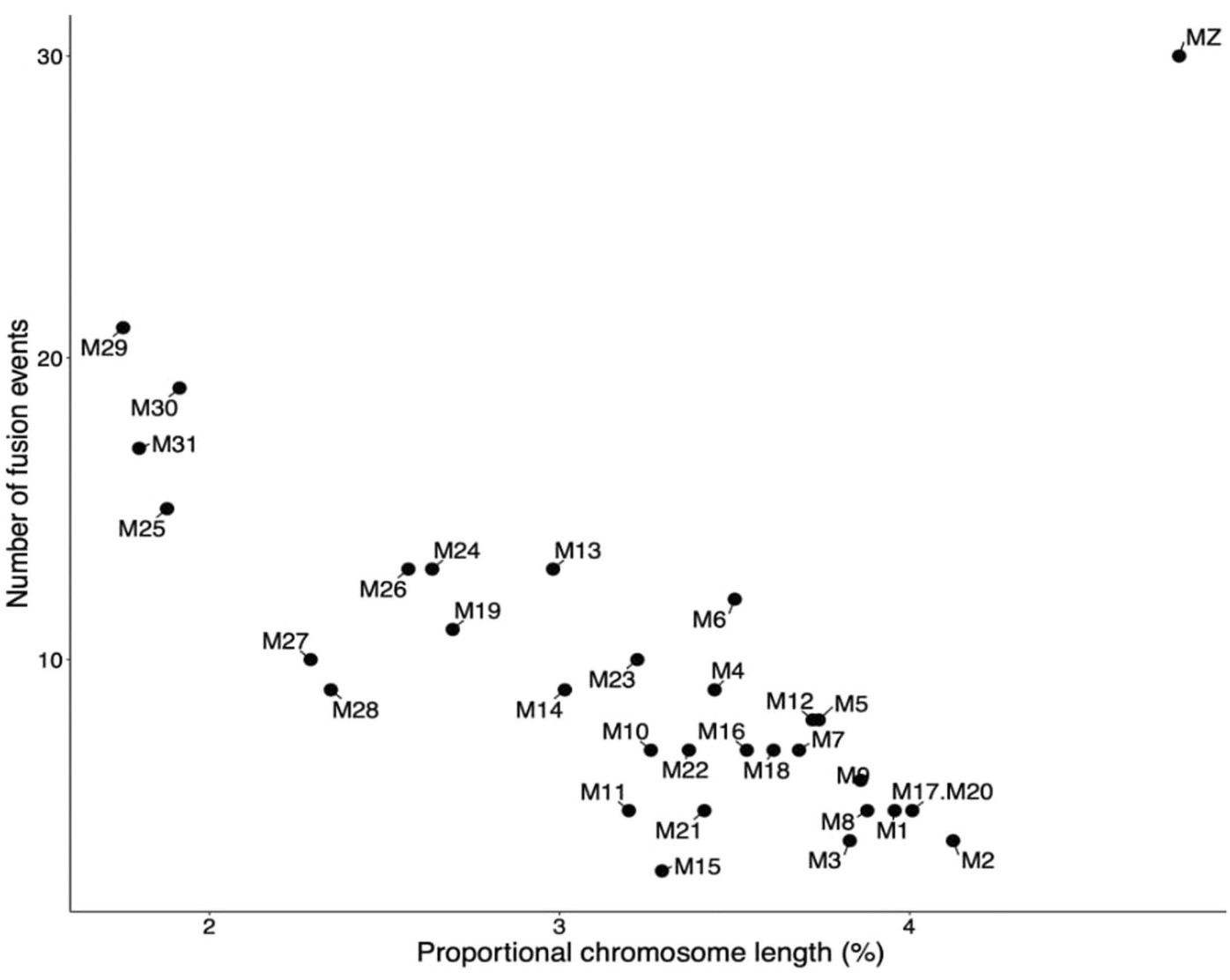

**Extended Data Fig. 3 | Relationship between proportional chromosome length and frequency of fusion events.** Number of fusion events that each Merian element is involved in compared to the average proportional length of the Merian element in the 210 species. Proportional chromosome length is chromosome length divided by genome size.

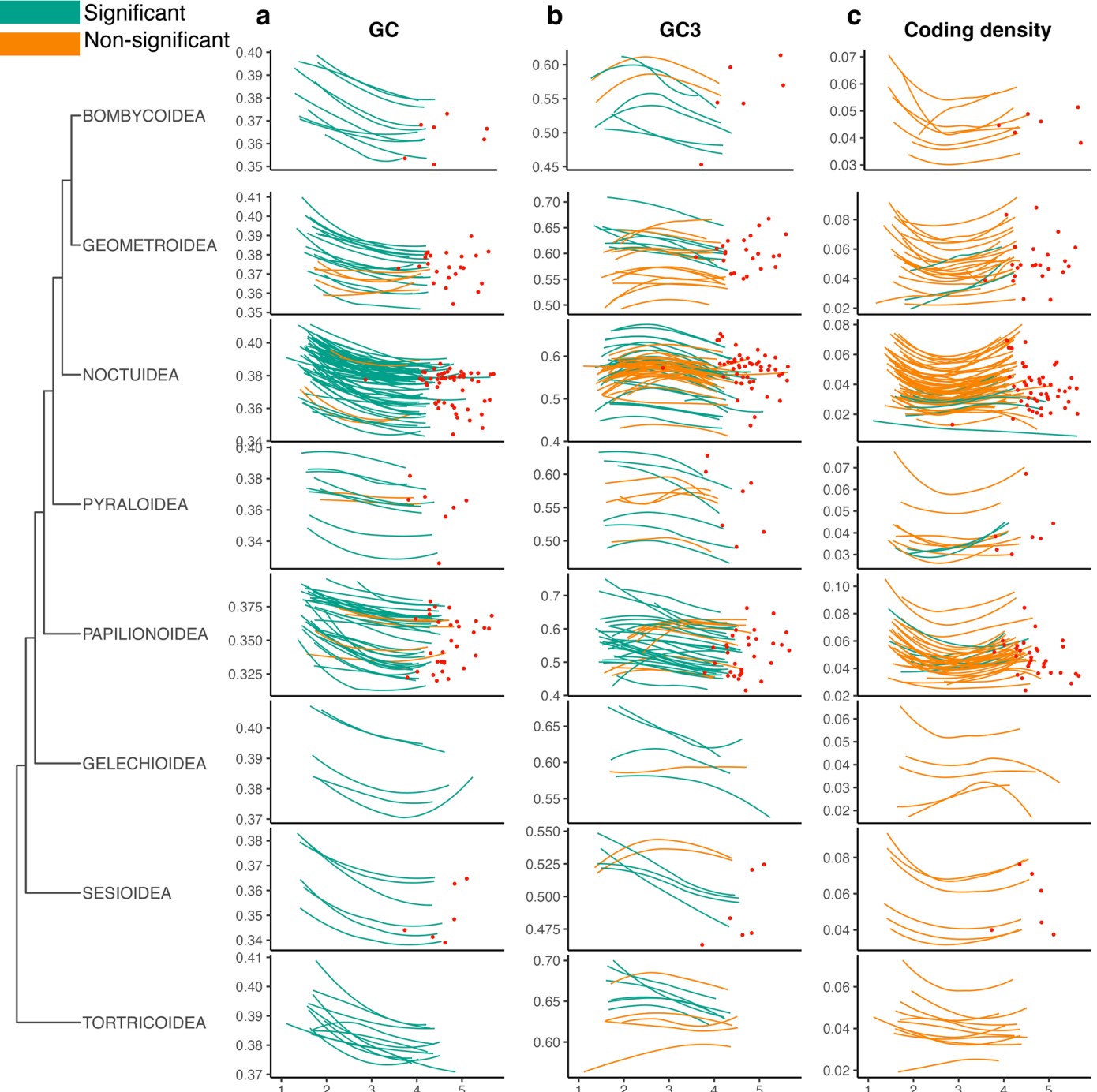

**Extended Data Fig. 4 | Correlation of GC, GC3 and coding density with chromosome length in Lepidoptera superfamilies. a–c**, Proportional chromosome length against GC proportion (**a**), GC proportion at the third codon position (GC3) (**b**), density of coding sequence (**c**). Proportional chromosome length is chromosome length divided by genome size. A locally weighted smoothing line ('LOESS') is drawn between the autosomes of each species. The line is coloured green if the correlation was significant (Spearman's rank, p < 0.05), or orange if it was non-significant (Supplementary Table 8). Only species with at least 10 autosomes are included and all autosomes that we inferred to have undergone fusion or fission events were removed. Only superfamilies represented by at least 5 superfamilies are shown. The Z chromosomes were not included in the correlation analysis and are indicated in red.

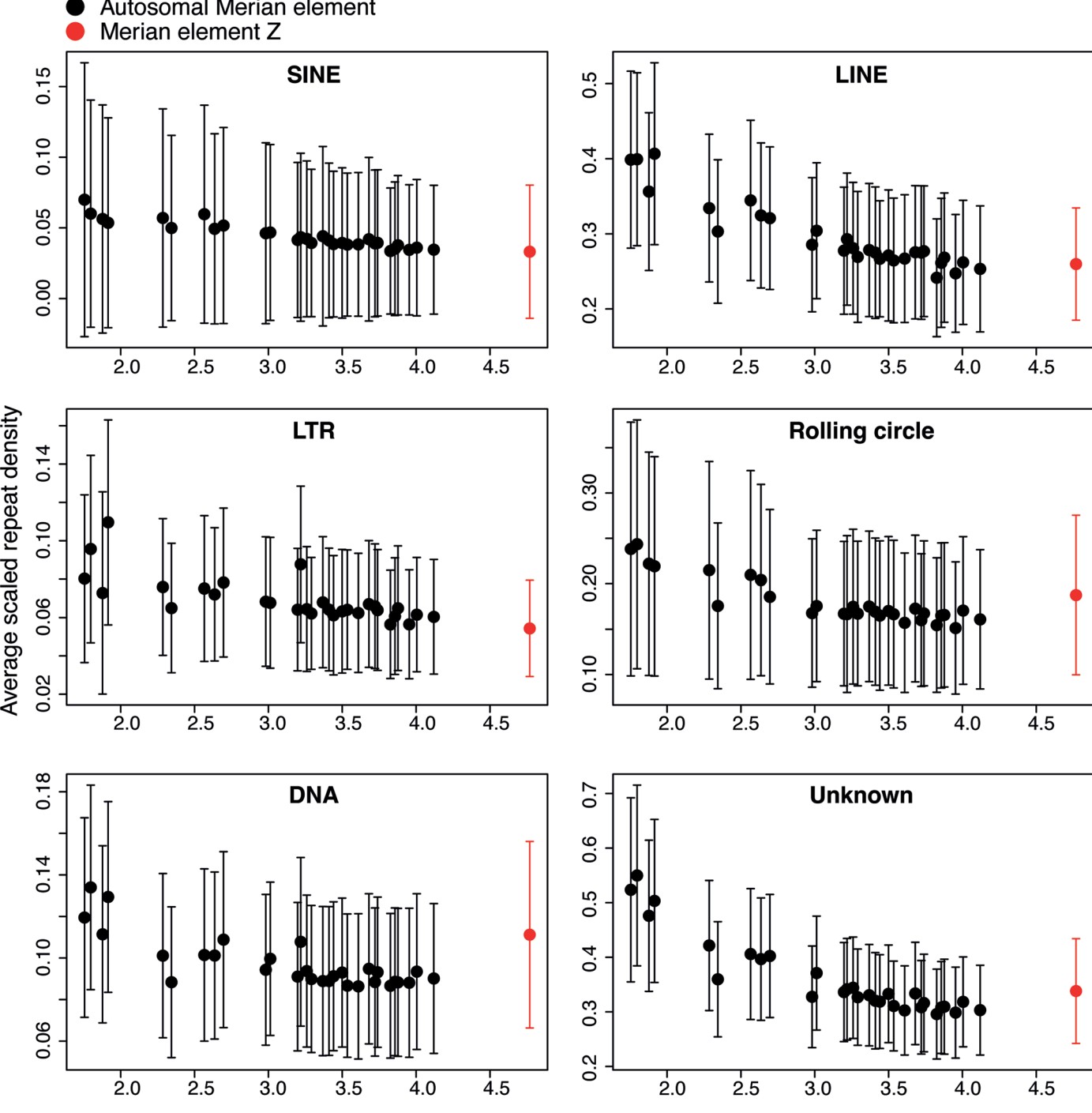

**Extended Data Fig. 5 | Relationship between repetitive element density and proportional chromosome length for each major class of transposable elements.** Mean proportional chromosome length (length divided by genome size) of each Merian element compared to the mean density of each repetitive element class. To enable comparison between species with different average repeat densities, the density of each repetitive element class was scaled by the mean repeat density of the genome. Only chromosomes that had not undergone fusions or fission events are included. MZ is indicated in red. Mean and standard deviation are shown. 5,471 chromosomes were examined and the number of chromosomes per Merian element ranged from 142 to 191, with a mean of 176 (Supplementary Tables 9, 10).

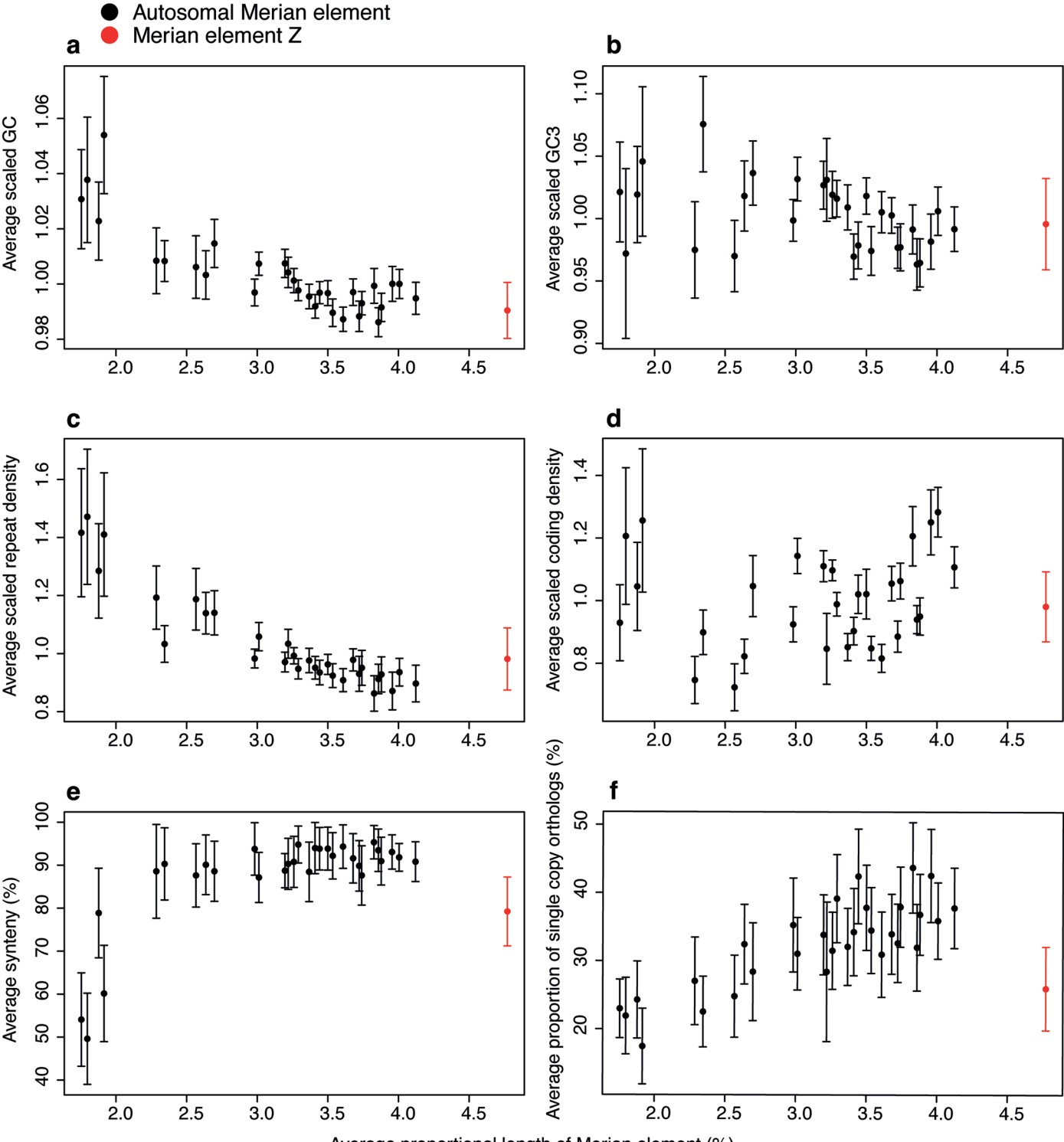

**Extended Data Fig. 6 | Relationship between GC, GC3, repeat density, coding density, synteny and proportion of single-copy orthologues with the proportional chromosome length of each Merian element. a, f,** Mean proportional chromosome length (length divided by genome size) compared to mean GC content of the Merian element in each genome scaled by the mean GC content of the genome (**a**), mean GC3 content off the Merian element in each genome scaled by the mean GC3 content of the genome (**b**), compared to mean repeat density of the Merian element in each genome scaled by the mean repeat density of the genome (**c**), coding density of the Merian element scaled by mean coding density of the genome (**d**), mean level of synteny of the Merian element (**e**), mean proportion of genes on the chromosome that are single copy and present in the majority of species (**f**). Only chromosomes that have not undergone fusions or fission events were included. MZ is indicated in red. Mean and standard deviation are shown. For GC, repeat density and synteny, 5,471 chromosomes were examined and the number of chromosomes per Merian element ranged from 142 to 191 with a mean of 176 (Supplementary Tables 9, 10). For GC3, coding density and the proportion of single-copy orthologues, a total of 5199 chromosomes were examined and the number of chromosomes per Merian element ranged from 134 to 182 with a mean of 168 (Supplementary Tables 9, 10).

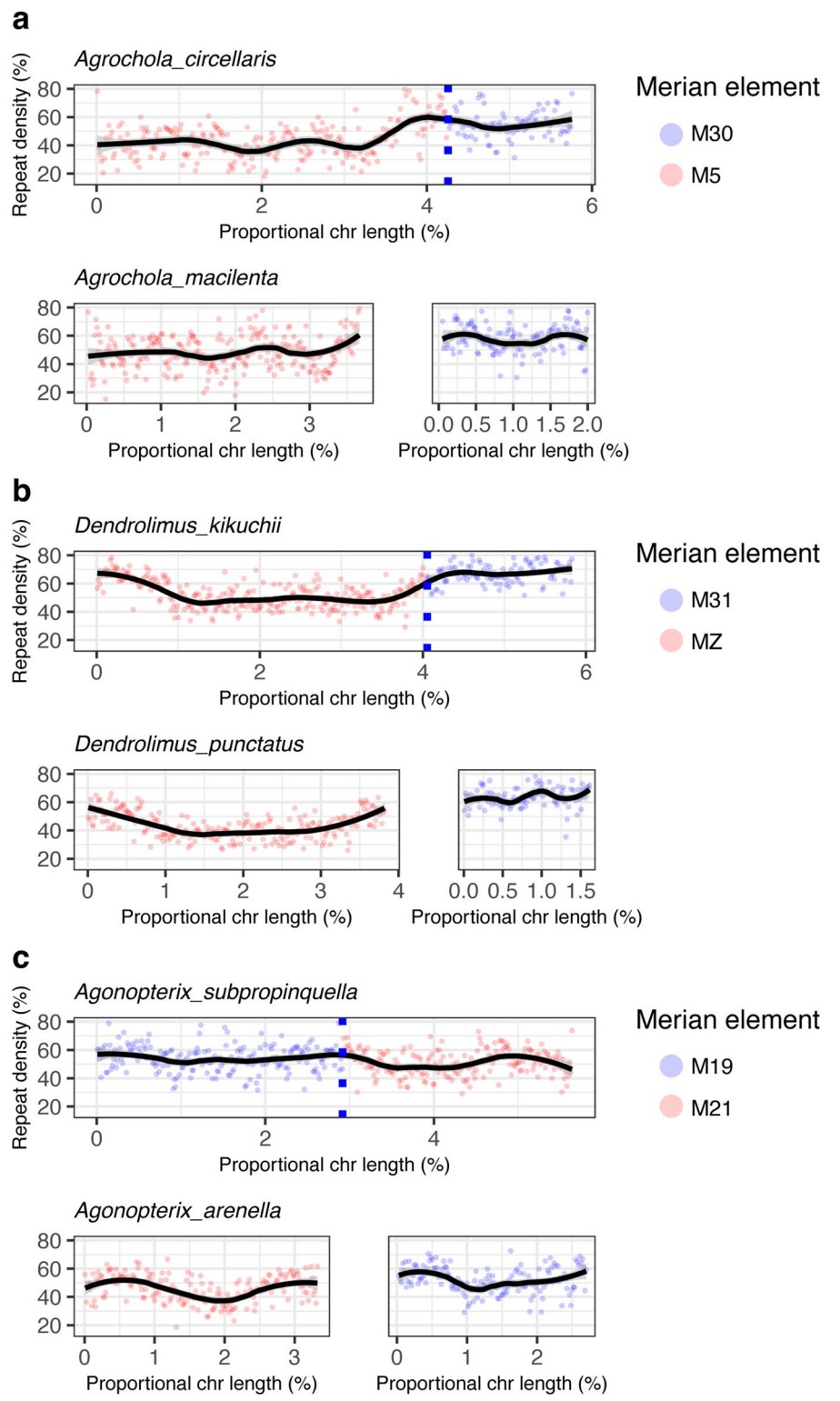

**Extended Data Fig. 7 | Repeat density across fused chromosomes and their unfused homologues in pairs of sister species. a–c**, Repeat density in fused chromosomes in one species, compared to the repeat density of the unfused orthologous chromosomes in a sister species. As each fusion is present in one species and absent in the other sampled species from the same genus, they are likely very recent fusions. *Agrochola circellaris*, compared to *A. macilenta*

(**a**), *Dendrolimus kikuchii* compared to *D. punctatus* (**b**) and *Agonopterix subpropinquella* and *A. arenella* (**c**). Repeat density is plotted along each chromosome in 100 kb windows, where the chromosome position is scaled to proportional length by dividing by genome size. Lines represent LOESS smoothing functions fitted to the data. Points are coloured by Merian element. Blue dashed lines indicate the fusion point along fused chromosomes.

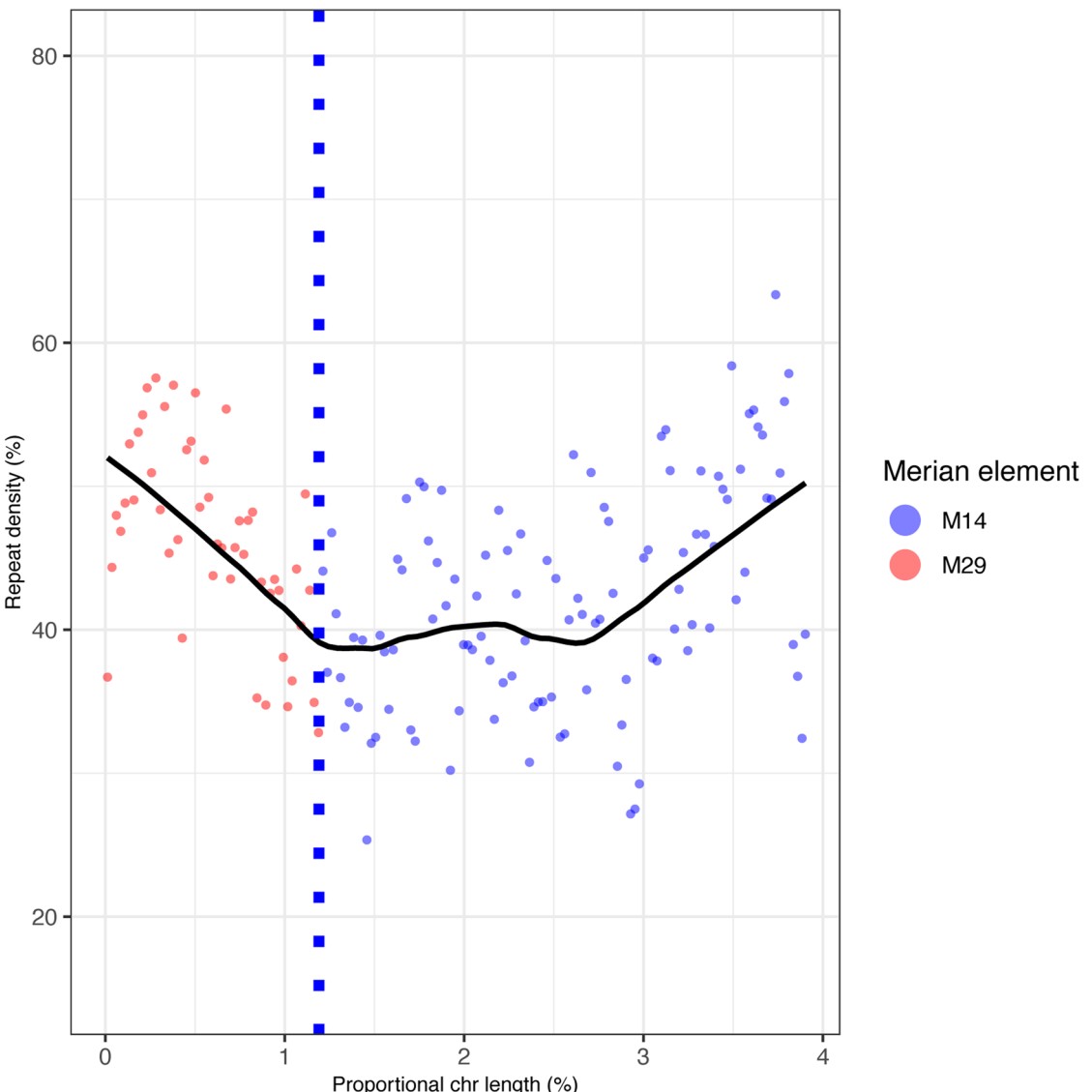

**Extended Data Fig. 8 | Repeat density across the fused chromosome in *Aphantopus hyperantus*.** Repeat density across the fused chromosome (M29 + M14) in *Aphantopus hyperantus*. The distance between *A. hyperantus* and the closest relative with an unfused M29 and M14 (*Danaus plexippus*) is too large for the unfused chromosomes to be a suitable proxy for the ancestral chromosomes and so is not shown. Repeat density is plotted along the chromosome in 100 kb windows, where the chromosome position is scaled to proportional length by dividing by genome size. Lines represent LOESS smoothing functions fitted to the data. Points are coloured by Merian element. Blue dashed lines indicate the fusion point along the fused chromosome.

# Reporting Summary

## Statistics

For all statistical analyses, confirm that the following items are present in the figure legend, table legend, main text, or Methods section.

| n/a | Confirmed | |
|---|---|---|
| ☐ | ☒ | The exact sample size (*n*) for each experimental group/condition, given as a discrete number and unit of measurement |
| ☐ | ☒ | A statement on whether measurements were taken from distinct samples or whether the same sample was measured repeatedly |
| ☐ | ☒ | The statistical test(s) used AND whether they are one- or two-sided *Only common tests should be described solely by name; describe more complex techniques in the Methods section.* |
| ☐ | ☒ | A description of all covariates tested |
| ☒ | ☐ | A description of any assumptions or corrections, such as tests of normality and adjustment for multiple comparisons |
| ☐ | ☒ | A full description of the statistical parameters including central tendency (e.g. means) or other basic estimates (e.g. regression coefficient) AND variation (e.g. standard deviation) or associated estimates of uncertainty (e.g. confidence intervals) |
| ☐ | ☒ | For null hypothesis testing, the test statistic (e.g. *F*, *t*, *r*) with confidence intervals, effect sizes, degrees of freedom and *P* value noted *Give P values as exact values whenever suitable.* |
| ☒ | ☐ | For Bayesian analysis, information on the choice of priors and Markov chain Monte Carlo settings |
| ☐ | ☒ | For hierarchical and complex designs, identification of the appropriate level for tests and full reporting of outcomes |
| ☐ | ☒ | Estimates of effect sizes (e.g. Cohen's *d*, Pearson's *r*), indicating how they were calculated |

*Our web collection on statistics for biologists contains articles on many of the points above.*

## Software and code

Policy information about availability of computer code

| | |
|---|---|
| Data collection | No software was used to collect data. |
| Data analysis | All custom code developed for this manuscript is available at https://github.com/charlottewright/Chromosome_evolution_Lepidoptera_MS which has been accessioned in the Zenodo repository https://doi.org/10.5281/zenodo.10373060.<br>All third-party software used in the manuscript, including versions, is detailed in the methods along with relevant citations and listed below.<br><br>Third-party software and versions used in the manuscript:<br>Earl Grey (v1.2)<br>BUSCO (v.5.4.3)<br>MAFFT (v7.475)<br>trimal (v1.4)<br>IQ-TREE (v2.03)<br>phylolm R package (v2.6.2)<br>BEDtools (v2.30.0)<br>AGAT (v1.0.0)<br>samtools (v1.7)<br>OrthoFinder (v2.5.4).<br>Phylolm (2.6.2)<br>ete3 (3.1.3)<br>stats R package (4.1.0)<br>fasta_windows (0.2.4) |

The following software tools are not versioned but are available on GitHub, as indicated in the methods section of the manuscript: busco2fasta, catfasta2phyml, gff-stats, genomics_tools, syngraph, lep_fusion_fission_finder, lep_busco_painter, assemblage.

For manuscripts utilizing custom algorithms or software that are central to the research but not yet described in published literature, software must be made available to editors and reviewers. We strongly encourage code deposition in a community repository (e.g. GitHub). See the Nature Portfolio guidelines for submitting code & software for further information.

# Data

Policy information about availability of data

All manuscripts must include a data availability statement. This statement should provide the following information, where applicable:
- Accession codes, unique identifiers, or web links for publicly available datasets
- A description of any restrictions on data availability
- For clinical datasets or third party data, please ensure that the statement adheres to our policy

The reference genomes analysed in this study are available at https://www.ncbi.nlm.nih.gov/ and the accession numbers are given in Supplementary Table 1. Gene annotations are available at Rapid.ensembl.org and are listed in Supplementary Table 3. The Arthropoda library from Dfam release 3.5  used to identify transposable elements is available at https://www.dfam.org/releases/Dfam_3.5. Large data files associated with this manuscript, including repeat annotations, repeat libraries and phylogenies are available in the https://doi.org/10.5281/zenodo.7925505 [https://zenodo.org/doi/10.5281/zenodo.7925505]. Other data that supports the findings presented in this paper are available in the Supplementary Tables and on GitHub https://github.com/charlottewright/Chromosome_evolution_Lepidoptera_MS which has been accessioned in Zenodo at https://doi.org/10.5281/zenodo.10373060. Additional source data associated with the figures and tables can be found in the Source Data file.

# Research involving human participants, their data, or biological material

Policy information about studies with human participants or human data. See also policy information about sex, gender (identity/presentation), and sexual orientation and race, ethnicity and racism.

| | |
|---|---|
| Reporting on sex and gender | N/A |
| Reporting on race, ethnicity, or other socially relevant groupings | N/A |
| Population characteristics | N/A |
| Recruitment | N/A |
| Ethics oversight | N/A |

Note that full information on the approval of the study protocol must also be provided in the manuscript.

# Field-specific reporting

Please select the one below that is the best fit for your research. If you are not sure, read the appropriate sections before making your selection.

☒ Life sciences          ☐ Behavioural & social sciences          ☐ Ecological, evolutionary & environmental sciences

For a reference copy of the document with all sections, see nature.com/documents/nr-reporting-summary-flat.pdf

# Life sciences study design

All studies must disclose on these points even when the disclosure is negative.

| | |
|---|---|
| Sample size | The analyses performed in this manuscript are based on a sample size of 210 lepidopteran genomes and 4 trichopteran genomes. This sampling was determined by the availability of chromosome-level reference genomes on INSDC on 27th June 2022. |
| Data exclusions | Two chromosome-level reference genomes were excluded from our analysis due to quality issues. The first, Zerene cesonia (GCA 012273895.2), contained 246 unlocalised scaffolds that contained 351 BUSCOs. The high number of BUSCOs in these scaffolds means that erroneous rearrangement events would be inferred if this genome were to be included. In the second, Cnaphalocrocis medinalis (GCA 014851415.1), the majority of genes belonging to the M30 Merian element were present on unlocalised scaffolds. We identified two additional genomes that contained minor misassembly issues that we were able to address prior to downstream analysis. In Dendrolimus kikuchii (GCA 019925095.1), we found three scaffolds with a high proportion of duplicated BUSCOs (the majority of which corresponded to the M30 Merian element), indicating that they represented haplotypic duplication. When we removed these scaffolds from the assembly, we successfully recovered a fusion between M30 and MZ that would have otherwise been missed. In Spodoptera frugiperda (GCA 011064685.2), we removed an unlocalised scaffold that contained 22 BUSCOs prior to downstream analyses to avoid inferring a fission event in this species due to assembly issues. |
| Replication | Our findings were replicated by performing two independent analyses that arrived at the same results for the inference of chromosome |

| Replication | rearrangements. Each analysis was performed once as replication was not relevant. Otherwise, as this was a comparative genomic study which drew upon all publicly-available chromosome-level reference genomes for Lepidoptera and Trichoptera (with the exception of the two genomes described above) and so using replicates was not relevant. |
|---|---|
| Randomization | Randomisation was not relevant to this study as we analysed all publicly-available chromosome-level reference genomes for Lepidoptera and Trichoptera (with the exception of the two genomes described above). |
| Blinding | As above, blinding was not relevant to this study as we analysed all publicly-available chromosome-level reference genomes for Lepidoptera and Trichoptera (with the exception of the two genomes described above). Blinding is also not relevant as we did not carry out a randomised control trial. |

# Reporting for specific materials, systems and methods

We require information from authors about some types of materials, experimental systems and methods used in many studies. Here, indicate whether each material, system or method listed is relevant to your study. If you are not sure if a list item applies to your research, read the appropriate section before selecting a response.

## Materials & experimental systems

| n/a | Involved in the study |
|---|---|
| ☒ | Antibodies |
| ☒ | Eukaryotic cell lines |
| ☒ | Palaeontology and archaeology |
| ☒ | Animals and other organisms |
| ☒ | Clinical data |
| ☒ | Dual use research of concern |
| ☒ | Plants |

## Methods

| n/a | Involved in the study |
|---|---|
| ☒ | ChIP-seq |
| ☒ | Flow cytometry |
| ☒ | MRI-based neuroimaging |

