## [Peer Review File · Nature Ecology & Evolution]

Peer Review Information

Journal: Nature Ecology & Evolution

Manuscript Title: Comparative genomics reveals the dynamics of chromosome evolution in
Lepidoptera

Corresponding author name(s): Charlotte J. Wright, Mark Blaxter

Editorial Notes:

Reviewer Comments & Decisions:

Decision Letter, initial version:

24th November 2023

Dear Charlotte,

Thank you for submitting your revised manuscript "Chromosome evolution in Lepidoptera" (NATECOLEVOL-23102390-T). It has now been seen again by the original reviewers and their comments are below. The reviewers find that the paper has improved in revision, and therefore we'll be happy in principle to publish it in Nature Ecology & Evolution, pending minor revisions to satisfy the reviewers' final requests and to comply with our editorial and formatting guidelines.

[REDACTED]

Reviewer #1 (Remarks to the Author):

We would like to thank the authors for their work on our comments. Overall, the paper has been improved and became more balanced, in particular via the addition of an additional chromosomal reconstruction method and bootstrap attempts. Here are a few remaining minor items that could still help improve your work:

- The comparison to "bilaterian" linkage group is a good addition, but from the Supplementary Figure 7 it is not correctly to say that the Merian elements are "highly rearranged". In fact, this Figure shows clear correspondence between the so-called "BLG"s and the Merian elements. In that case, it would be best to also show a dotplot of ortholog distribution along the chromosomes. This will enable the readers to see how exactly the orthogroups are distributed on lepidopteran chromosomes, especially those that have multiple more ancestral chromosomal element contributions. This will help elaborate the new statements in the text such as "Merian elements are distinct from, but retain signatures consistent with, the BLGs (Supplementary Fig. 7)." What does this mean, are they distinct or show retained signatures? If they show retained signatures, which BLGs contribute to which Merian elements?

- Thank you for discussing further the "breakage" of Merian elements. Please elaborate further at what

2degree of “breakage” would you consider giving new names to the rearranged chromosomes? Would it warrant calling them neo-Merian elements, or are these rearrangements too few? It is perhaps obvious, by also a short explanation as to why you call these linkages “Merian elements” might be useful.

- In respect to the observation that deeper nodes show less fusions, you should probably mention that this might be primarily affected by the species sampling and evolutionary rates within each taxon that you study, as such the “uniform distribution” might not be the right null expectation.

Reviewer #2 (Remarks to the Author):

I have carefully reviewed the responses to reviewers and the revised manuscript. I find that the authors have satisfactorily addressed my comments. As mentioned in my previous review, I believe this paper presents an important step forward in using comparative genomics to examine insect chromosome evolution and I think the revised manuscript is much improved.

Reviewer #3 (Remarks to the Author):

Dear Authors,

I have now had a chance to read through your response to reviewers, as part of your submission to NEE. I greatly appreciate your efforts to conduct more thorough assessments of your Merian elements, and their evolutionary history, in response to diverse reviewer concerns. I find the revised manuscript improved and additional analyses helpful in several ways. At this point my only concern resides in what I consider to be a lingering overstatement of the novelty of your findings.

In your response to Rev1 concerns about the role of smaller chromosomes in fusion events, you revise your main text to state “our discovery of small chromosomes driving evolutionary change in genome organisation in Lepidoptera shows some similarity to the monocentric chromosomes of vertebrates”. Here, you are overstating your findings, which I find surprising, as you are clearly aware that such “discoveries” have been previously published (e.g. Ahola et al. 2014). Where is the nuanced assessment of literature that you nice display on lines L283-287,

“A bias towards the involvement of smaller chromosomes in fusion events in evolutionary distant lineages has been suggested previously based on analysis of fusions in *B. mori* and *H. melpomene* compared to *M. cinxia*43. Our analysis suggests that this holds across Lepidoptera and is true for both autosome-autosome fusions and Z-autosome fusions.”

In sum, your work does not “discover” that small fusions drive evol change, but rather refines previous discoveries by documenting the extent of a previously described pattern from both butterflies and moths. Your work provides tempo and mode, on a larger and better scale. Overstating such findings as “discoveries” detracts rather than adds to your work.

2Our ref: NATECOLEVOL-23102390-T

29th November 2023

Dear Dr. Wright,

Thank you for your patience as we've prepared the guidelines for final submission of your Nature Ecology & Evolution manuscript, "Chromosome evolution in Lepidoptera" (NATECOLEVOL-23102390-T). Please carefully follow the step-by-step instructions provided in the attached file, and add a response in each row of the table to indicate the changes that you have made. Please also check and comment on any additional marked-up edits we have proposed within the text. Ensuring that each point is addressed will help to ensure that your revised manuscript can be swiftly handed over to our production team.

****We would like to start working on your revised paper, with all of the requested files and forms, as soon as possible (preferably within two weeks). Please get in contact with us immediately if you anticipate it taking more than two weeks to submit these revised files.****

In recognition of the time and expertise our reviewers provide to Nature Ecology & Evolution's editorial process, we would like to formally acknowledge their contribution to the external peer review of your manuscript entitled "Chromosome evolution in Lepidoptera". For those reviewers who give their assent, we will be publishing their names alongside the published article.

Nature Ecology & Evolution offers a Transparent Peer Review option for new original research manuscripts submitted after December 1st, 2019. As part of this initiative, we encourage our authors to support increased transparency into the peer review process by agreeing to have the reviewer comments, author rebuttal letters, and editorial decision letters published as a Supplementary item. When you submit your final files please clearly state in your cover letter whether or not you would like to participate in this initiative. Please note that failure to state your preference will result in delays in accepting your manuscript for publication.

Cover suggestions

3We welcome submissions of artwork for consideration for our cover. For more information, please see our [guide for cover artwork](https://www.nature.com/documents/Nature_covers_author_guide.pdf).

Nature Ecology & Evolution has now transitioned to a unified Rights Collection system which will allow our Author Services team to quickly and easily collect the rights and permissions required to publish your work. Approximately 10 days after your paper is formally accepted, you will receive an email in providing you with a link to complete the grant of rights. If your paper is eligible for Open Access, our Author Services team will also be in touch regarding any additional information that may be required to arrange payment for your article.

Please note that *Nature Ecology & Evolution* is a Transformative Journal (TJ). Authors may publish their research with us through the traditional subscription access route or make their paper immediately open access through payment of an article-processing charge (APC). Authors will not be required to make a final decision about access to their article until it has been accepted. [Find out more about Transformative Journals](https://www.springernature.com/gp/open-research/transformative-journals)

Authors may need to take specific actions to achieve [compliance with funder and institutional open access mandates](https://www.springernature.com/gp/open-research/funding/policy-compliance-faqs). If your research is supported by a funder that requires immediate open access (e.g. according to [Plan S principles](https://www.springernature.com/gp/open-research/plan-s-compliance)) then you should select the gold OA route, and we will direct you to the compliant route where possible. For authors selecting the subscription publication route, the journal's standard licensing terms will need to be accepted, including <https://www.nature.com/nature-portfolio/editorial-policies/self-archiving-and-license-to-publish>. Those licensing terms will supersede any other terms that the author or any third party may assert apply to any version of the manuscript.

Please use the following link for uploading these materials:
[REDACTED]

4If you have any further questions, please feel free to contact me.

[REDACTED]

Reviewer #1:

Remarks to the Author:

We would like to thank the authors for their work on our comments. Overall, the paper has been improved and became more balanced, in particular via the addition of an additional chromosomal reconstruction method and bootstrap attempts. Here are a few remaining minor items that could still help improve your work:

- The comparison to "bilaterian" linkage group is a good addition, but from the Supplementary Figure 7 it is not correctly to say that the Merian elements are "highly rearranged". In fact, this Figure shows clear correspondence between the so-called "BLG"s and the Merian elements. In that case, it would be best to also show a dotplot of ortholog distribution along the chromosomes. This will enable the readers to see how exactly the orthogroups are distributed on lepidopteran chromosomes, especially those that have multiple more ancestral chromosomal element contributions. This will help elaborate the new statements in the text such as "Merian elements are distinct from, but retain signatures consistent with, the BLGs (Supplementary Fig. 7)." What does this mean, are they distinct or show retained signatures? If they show retained signatures, which BLGs contribute to which Merian elements?
- Thank you for discussing further the "breakage" of Merian elements. Please elaborate further at what degree of "breakage" would you consider giving new names to the rearranged chromosomes? Would it warrant calling them neo-Merian elements, or are these rearrangements too few? It is perhaps obvious, by also a short explanation as to why you call these linkages "Merian elements" might be useful.
- In respect to the observation that deeper nodes show less fusions, you should probably mention that this might be primarily affected by the species sampling and evolutionary rates within each taxon that you study, as such the "uniform distribution" might not be the right null expectation.

Reviewer #2:

Remarks to the Author:

I have carefully reviewed the responses to reviewers and the revised manuscript. I find that the authors have satisfactorily addressed my comments. As mentioned in my previous review, I believe this paper presents an important step forward in using comparative genomics to examine insect chromosome evolution and I think the revised manuscript is much improved.

Reviewer #3:

Remarks to the Author:

Dear Authors,

5I have now had a chance to read through your response to reviewers, as part of your submission to NEE. I greatly appreciate your efforts to conduct more thorough assessments of your Merian elements, and their evolutionary history, in response to diverse reviewer concerns. I find the revised manuscript improved and additional analyses helpful in several ways. At this point my only concern resides in what I consider to be a lingering overstatement of the novelty of your findings.

In your response to Rev1 concerns about the role of smaller chromosomes in fusion events, you revise your main text to state “our discovery of small chromosomes driving evolutionary change in genome organisation in Lepidoptera shows some similarity to the monocentric chromosomes of vertebrates”. Here, you are overstating your findings, which I find surprising, as you are clearly aware that such “discoveries” have been previously published (e.g. Ahola et al. 2014). Where is the nuanced assessment of literature that you nice display on lines L283-287,

“A bias towards the involvement of smaller chromosomes in fusion events in evolutionary distant lineages has been suggested previously based on analysis of fusions in *B. mori* and *H. melpomene* compared to *M. cinxia*⁴³. Our analysis suggests that this holds across Lepidoptera and is true for both autosome-autosome fusions and Z-autosome fusions.”

In sum, your work does not “discover” that small fusions drive evol change, but rather refines previous discoveries by documenting the extent of a previously described pattern from both butterflies and moths. Your work provides tempo and mode, on a larger and better scale. Overstating such findings as “discoveries” detracts rather than adds to your work.

Author Rebuttal to Initial comments

Referee 1

We would like to thank the authors for their work on our comments. Overall, the paper has been improved and became more balanced, in particular via the addition of an additional chromosomal reconstruction method and bootstrap attempts. Here are a few remaining minor items that could still help improve your work:

We thank the reviewer for taking the time to assess our revised manuscript. We are glad they found the work improved. In the responses below, we discuss the reviewers' suggestions.

- The comparison to “bilaterian” linkage group is a good addition, but from the Supplementary Figure 7 it is not correctly to say that the Merian elements are “highly rearranged”. In fact, this Figure shows clear

6correspondence between the so-called “BLG”s and the Merian elements. In that case, it would be best to also show a dotplot of ortholog distribution along the chromosomes. This will enable the readers to see how exactly the orthogroups are distributed on lepidopteran chromosomes, especially those that have multiple more ancestral chromosomal element contributions. This will help elaborate the new statements in the text such as “Merian elements are distinct from, but retain signatures consistent with, the BLGs (Supplementary Fig. 7).” What does this mean, are they distinct or show retained signatures? If they show retained signatures, which BLGs contribute to which Merian elements?

As expected based on common ancestry, Merian elements and BLGs overlap in their orthologue composition. However, rearrangement has occurred relative to BLGs, meaning that each Merian element has contributions from multiple BLGs. On average, the highest proportion of each Merian element in a BLG is just 40%, highlighting that BLGs are fragmented across Merian elements.

We agree with the reviewer that our description of this could be clearer in the main text. We have adjusted it to read:

“We compared the distribution of the orthologues allocated to Merian elements to their allocation to bilaterian ancestral linkage groups (BLGs; $n=24$)⁷, from which Merian elements descend from, and which date to approximately 560 million years ago⁴⁸. As expected, Merian elements show some correspondence to BLGs, with seventeen Merian elements showing greater similarity in orthologue assignment with BLGs than expected under random sampling. However, most Merian elements were rearranged relative to BLGs, possessing combinations of loci from multiple BLGs (Extended Data Fig. 2a-c).” (lines 181-190)

The proportion of BLG orthologues that were assigned to each Merian element is shown in the heatmap of Supplementary Figure 7.

Although we agree that it would be interesting to compare BLGs with Merian elements using dot plots, this is not possible because gene orders for both BLGs and Merian elements have not been inferred. Instead, BLGs and Merian elements are defined as sets of genes without respect to gene order.

- Thank you for discussing further the “breakage” of Merian elements. Please elaborate further at what degree of “breakage” would you consider giving new names to the rearranged chromosomes? Would it warrant calling them neo-Merian elements, or are these rearrangements too few?

We would like to clarify to the reviewer that Merian elements are defined as the ancestral linkage groups of Lepidoptera and so describe the arrangement of genes across the chromosomes of the ancestral lepidopteran rather than names of chromosomes *per se*. We do not attempt to ‘name’ the chromosomes of any extant lepidopteran species, regardless of their level of rearrangement relative to Merian elements.

As the reviewer suggests, Merian elements have become fragmented in species that have undergone numerous fusions and fissions, such as in *Pieris*. To better understand these rearrangements, it would be possible to infer the ancestral linkage groups of the last common ancestor of each highly rearranged lineage. The tools that we employed, such as Syngraph, define the orthologue content of each internal node, meaning we can recover the ancestral linkage groups of each node. However, these would represent linkage groups of specific subsets of lepidopterans rather than of Lepidoptera overall, meaning that ‘neo-Merian elements’ might not be appropriate. Moreover, to accurately define the ancestral linkage groups in the rearranged ancestor of each highly-rearranged taxon would require deeper sampling within each lineage than present in our dataset.

It is perhaps obvious, by also a short explanation as to why you call these linkages “Merian elements” might be useful.

We agree that explaining why we named the lepidopteran linkage groups as ‘Merian elements’ is important. We do so in the main text:

“Hereafter, we refer to these ALGs as Merian elements, named after the seventeenth-century lepidopterist and botanical artist, Maria Sibylla Merian⁴⁵. (lines 136-137):

- In respect to the observation that deeper nodes show less fusions, you should probably mention that this might be primarily affected by the species sampling and evolutionary rates within each taxon that you study, as such the “uniform distribution” might not be the right null expectation.

We thank the reviewer for this suggestion. The fact that our results are not recovered by simulations that assume a uniform rate of rearrangement suggests a departure from this very simple model. One explanation for this departure is that chromosome rearrangements have an effect on rates of extinction and speciation (which we discuss in the main text). We agree with the reviewer that variation in evolutionary rates of rearrangement across the tree is another explanation, although it is not obvious why rates would be elevated on external branches. We note this in the discussion: ‘Alternative explanations, such as a general increase in the rate of fixation of fusions in recently evolutionary time or frequent reversion by exact fission seem unlikely.’ Given our dataset, where only a small number of families are densely sampled, we have not attempted to infer rates of rearrangements across different parts of the tree. To acknowledge the limitations in our sampling and analysis, we now write in the discussion:

“We note that this analysis is based on a fraction of Lepidopteran diversity, and requires deeper investigation with denser species sampling.” (lines 406-408)

Referee 2

I have carefully reviewed the responses to reviewers and the revised manuscript. I find that the authors have satisfactorily addressed my comments. As mentioned in my previous review, I believe this paper presents an important step forward in using comparative genomics to examine insect chromosome evolution and I think the revised manuscript is much improved.

We thank the reviewer for reading our revised manuscript and are glad that they find it to be improved.

Referee 3

Dear Authors,

9I have now had a chance to read through your response to reviewers, as part of your submission to NEE. I greatly appreciate your efforts to conduct more thorough assessments of your Merian elements, and their evolutionary history, in response to diverse reviewer concerns. I find the revised manuscript improved and additional analyses helpful in several ways. At this point my only concern resides in what I consider to be a lingering overstatement of the novelty of your findings.

We thank the reviewer for considering our revised manuscript. We are pleased that the reviewer finds the work has benefitted from the additional analyses.

In your response to Rev1 concerns about the role of smaller chromosomes in fusion events, you revise your main text to state “our discovery of small chromosomes driving evolutionary change in genome organisation in Lepidoptera shows some similarity to the monocentric chromosomes of vertebrates”. Here, you are overstating your findings, which I find surprising, as you are clearly aware that such “discoveries” have been previously published (e.g. Ahola et al. 2014). Where is the nuanced assessment of literature that you nice display on lines L283-287,

*“A bias towards the involvement of smaller chromosomes in fusion events in evolutionary distant lineages has been suggested previously based on analysis of fusions in *B. mori* and *H. melpomene* compared to *M. cinxia*⁴³. Our analysis suggests that this holds across Lepidoptera and is true for both autosome-autosome fusions and Z-autosome fusions.”*

In sum, your work does not “discover” that small fusions drive evol change, but rather refines previous discoveries by documenting the extent of a previously described pattern from both butterflies and moths. Your work provides tempo and mode, on a larger and better scale. Overstating such findings as “discoveries” detracts rather than adds to your work.

We agree with the reviewer that the word ‘discovery’ in this sentence is not the correct word. Accordingly, we have revised this sentence:

“Small Merian elements show some similarities to the monocentric, GC-rich, microchromosomes of vertebrates⁶¹. Interestingly, comparative analyses indicate that the ancestral vertebrates possessed a set of small gene-rich chromosomes. Subsequently, subsets of microchromosomes progressively fused, resulting in macrochromosomes. Therefore, our finding of the involvement of small chromosomes in genome reorganisation across Lepidoptera shows some similarity to vertebrate chromosome evolution.” (lines 427-435)

Additional minor changes

We have made a set of minor text changes to make the text more concise. As a result, the main text is now under 5,000 words. These changes, in addition to all changes in response to the reviewers' comments, are indicated in the Word document tracked changes.

Final Decision Letter:

12th January 2024

Dear Charlotte,

We are pleased to inform you that your Article entitled "Comparative genomics reveals the dynamics of chromosome evolution in Lepidoptera", has now been accepted for publication in Nature Ecology & Evolution.

Over the next few weeks, your paper will be copyedited to ensure that it conforms to Nature Ecology and Evolution style. Once your paper is typeset, you will receive an email with a link to choose the appropriate publishing options for your paper and our Author Services team will be in touch regarding any additional information that may be required

Due to the importance of these deadlines, we ask you please us know now whether you will be difficult to contact over the next month. If this is the case, we ask you provide us with the contact information

11(email, phone and fax) of someone who will be able to check the proofs on your behalf, and who will be available to address any last-minute problems. Once your paper has been scheduled for online publication, the Nature press office will be in touch to confirm the details.

Acceptance of your manuscript is conditional on all authors' agreement with our publication policies (see www.nature.com/authors/policies/index.html). In particular your manuscript must not be published elsewhere and there must be no announcement of the work to any media outlet until the publication date (the day on which it is uploaded onto our web site).

Please note that *Nature Ecology & Evolution* is a Transformative Journal (TJ). Authors may publish their research with us through the traditional subscription access route or make their paper immediately open access through payment of an article-processing charge (APC). Authors will not be required to make a final decision about access to their article until it has been accepted. [Find out more about Transformative Journals](https://www.springernature.com/gp/open-research/transformative-journals)

Authors may need to take specific actions to achieve [compliance with funder and institutional open access mandates](https://www.springernature.com/gp/open-research/funding/policy-compliance-faqs). If your research is supported by a funder that requires immediate open access (e.g. according to [Plan S principles](https://www.springernature.com/gp/open-research/plan-s-compliance)) then you should select the gold OA route, and we will direct you to the compliant route where possible. For authors selecting the subscription publication route, the journal's standard licensing terms will need to be accepted, including [self-archiving-and-license-to-publish](https://www.nature.com/nature-portfolio/editorial-policies/self-archiving-and-license-to-publish). Those licensing terms will supersede any other terms that the author or any third party may assert apply to any version of the manuscript.

We welcome the submission of potential cover material (including a short caption of around 40 words) related to your manuscript; suggestions should be sent to Nature Ecology & Evolution as electronic files (the image should be 300 dpi at 210 x 297 mm in either TIFF or JPEG format). Please note that such pictures should be selected more for their aesthetic appeal than for their scientific content, and that colour images work better than black and white or grayscale images. Please do not try to design a

12cover with the Nature Ecology & Evolution logo etc., and please do not submit composites of images related to your work. I am sure you will understand that we cannot make any promise as to whether any of your suggestions might be selected for the cover of the journal.

You can generate the link yourself when you receive your article DOI by entering it here: <http://authors.springernature.com/share>.

[REDACTED]

P.S. Click on the following link if you would like to recommend Nature Ecology & Evolution to your librarian <http://www.nature.com/subscriptions/recommend.html#forms>

** Visit the Springer Nature Editorial and Publishing website at http://editorial-jobs.springernature.com?utm_source=ejp_NEcoE_email&utm_medium=ejp_NEcoE_email&utm_campaign=ejp_NEcoE for more information about our career opportunities. If you have any questions please click [here](mailto:editorial.publishing.jobs@springernature.com). **